A novel chaotic transient search optimization algorithm for global optimization, real-world engineering problems and feature selection

http://orcid.org/0000-0003-3989-2432 Altay Osman osman.altay@cbu.edu.tr
Varol Altay Elif
Software Engineering, Manisa Celal Bayar University , Manisa , Turkey
Stević Željko
Electronic publication date: 2023 Aug 22
Publication date: 2023
Volume: 9
Electronic Location ID: e1526
Received 2023 Feb 2; Accepted 2023 Jul 19
Copyright: © 2023 Altay and Varol Altay
Copyright year: 2023
Copyright holder: Altay and Varol Altay
License: This is an open access article distributed under the terms of the Creative Commons Attribution License, which permits unrestricted use, distribution, reproduction and adaptation in any medium and for any purpose provided that it is properly attributed. For attribution, the original author(s), title, publication source (PeerJ Computer Science) and either DOI or URL of the article must be cited.
License URL: https://creativecommons.org/licenses/by/4.0/

Keywords: Chaotic transient search optimization algorithm, Chaotic maps, Benchmark functions, Real-world engineering problems, Feature selection

Funding: The authors received no funding for this work.

==============================
Metaheuristic optimization algorithms manage the search process to explore search domains efficiently and are used efficiently in large-scale, complex problems. Transient Search Algorithm (TSO) is a recently proposed physics-based metaheuristic method inspired by the transient behavior of switched electrical circuits containing storage elements such as inductance and capacitance. TSO is still a new metaheuristic method; it tends to get stuck with local optimal solutions and offers solutions with low precision and a sluggish convergence rate. In order to improve the performance of metaheuristic methods, different approaches can be integrated and methods can be hybridized to achieve faster convergence with high accuracy by balancing the exploitation and exploration stages. Chaotic maps are effectively used to improve the performance of metaheuristic methods by escaping the local optimum and increasing the convergence rate. In this study, chaotic maps are included in the TSO search process to improve performance and accelerate global convergence. In order to prevent the slow convergence rate and the classical TSO algorithm from getting stuck in local solutions, 10 different chaotic maps that generate chaotic values instead of random values in TSO processes are proposed for the first time. Thus, ergodicity and non-repeatability are improved, and convergence speed and accuracy are increased. The performance of Chaotic Transient Search Algorithm (CTSO) in global optimization was investigated using the IEEE Congress on Evolutionary Computation (CEC)’17 benchmarking functions. Its performance in real-world engineering problems was investigated for speed reducer, tension compression spring, welded beam design, pressure vessel, and three-bar truss design problems. In addition, the performance of CTSO as a feature selection method was evaluated on 10 different University of California, Irvine (UCI) standard datasets. The results of the simulation showed that Gaussian and Sinusoidal maps in most of the comparison functions, Sinusoidal map in most of the real-world engineering problems, and finally the generally proposed CTSOs in feature selection outperform standard TSO and other competitive metaheuristic methods. Real application results demonstrate that the suggested approach is more effective than standard TSO.

Introduction

Optimization is the process of identifying the most optimal solution to a problem from among all possible alternatives. Given the nature of optimization algorithms, they can be broadly classified into two categories: deterministic algorithms and stochastic intelligent algorithms. Additionally, stochastic algorithms are categorized into two types: heuristic algorithms and metaheuristic algorithms (Yang et al., 2012).

Deterministic optimization algorithms are insufficient for large-scale combinatorial and nonlinear problems. Usually, due to the natural solution mechanisms of deterministic algorithms, the problem of interest is modeled in such a way that the algorithm handles it. The solution strategy of deterministic methods usually depends on the types of objectives and constraints and the types of variables used in modeling the problem. The efficacy of these methods is significantly influenced by the solution space, the quantity of decision variables, and the number of constraints involved in problem formulation. An additional noteworthy limitation is the absence of overarching solution approaches that can be implemented for problem formulations featuring diverse decision objectives, variables, and constraints. That is, most algorithms solve models with certain types of objective functions or constraints. However, optimization problems in many different fields such as management science, computing, and engineering simultaneously require different types of decision variables, objective functions, and constraints in their formulation. Therefore, metaheuristic optimization algorithms have been proposed. These have become very popular methods in recent years because they have good computational power and are easy to transform (Bianchi et al., 2009).

Recently, metaheuristic algorithms have gained unexpected popularity. This is because they have demonstrated their superiority in tackling several optimization challenges. General-purpose metaheuristic methods are examined in different categories: biology-based, chemistry-based, mathematics-based, music-based, physics-based, plant-based, swarm-based, social-based, sports-based, water-based, and hybrid-based (Altay & Alatas, 2020). The emergence of physics-based algorithms was caused by physics phenomena in nature. The most well-known include Big Bang-Big Crunch (Erol & Eksin, 2006), electromagnetism-like heuristic (Birbil & Fang, 2003), central force optimization algorithm (Formato, 2009), multi verse optimization (MVO) (Mirjalili, Mirjalili & Hatamlou, 2016), galaxy-based search algorithm (Hosseini, 2011), Henry gas solubility optimization (Hashim et al., 2019), gradient-based optimizer (Ahmadianfar, Bozorg-Haddad & Chu, 2020), equilibrium optimizer (Faramarzi et al., 2020), flow direction algorithm (Karami et al., 2021), Archimedes optimization algorithm (Hashim et al., 2021), transit search algorithm (Mirrashid & Naderpour, 2022), and transient search algorithm (TSO) (Qais, Hasanien & Alghuwainem, 2020).

Metaheuristic methods are being developed thanks to their simplicity, cheap computational cost, gradient-free mechanism, and flexibility, and the interest in the use of these methods is increasing day by day. In this area, there is a theorem called No Free Lunch (NFL), which proves that there is no general algorithm for solving all optimization problems and allows this work area to be used actively. It has been mathematically proven by the NFL theorem that there is no single optimization method that solves all optimization algorithms. Thus, while the optimization algorithm produces good results in solving one problem, it can produce bad results in another. The gap here also encourages researchers working in this field to produce new methods, improve existing methods, or hybridize methods by combining them. Another gap is that, due to the stochastic optimization process, it is very difficult to maintain a balance between exploration and exploitation in the creation of any metaheuristic algorithm. TSO is a very new physics-based method inspired by the transient behavior of switched electrical circuits containing storage elements such as inductance and capacitance. As with other metaheuristic methods, TSO faces the challenge of achieving the right balance between exploration and exploitation. This study focuses on finding a solution to this problem, improving the speed of convergence and the ability of TSO to find the global optimum solution, and ultimately improving the performance of TSO in terms of various metrics. To the best of our knowledge, no studies have been done on how to increase global convergence and performance rates while avoiding trapping TSOs in local solutions. For the first time, chaos theory has been applied to TSO in this work to eliminate the drawbacks of the method. In this work, chaotic maps are embedded inside TSO to create novel algorithms known as chaotic TSO algorithms.

In this study, 10 different chaotic maps were integrated into the TSO algorithm. The main motivation for the study is to use the number sequences obtained from different chaotic maps instead of the critical parameters produced by random numbers in the TSO algorithm. By using chaotic maps with ergodic, irregular, and stochastic features in chaotic map TSO, it is aimed at avoiding local solutions more easily compared to the TSO method. In this way, it is aimed to increase global convergence and obtain a better curve by improving the exploration and exploitation stages of the TSO algorithm. The proposed method has been applied to the accepted CEC’17 benchmark functions, real-world engineering design problems, and feature selection in the literature.

The remainder of the article is organized as follows: In the second section, the working principle of the TSO algorithm is given. In the third section, chaotic maps are examined, and their equations are given. In the fourth section, the proposed chaotic TSO method is explained in detail. In the fifth section, the experimental results are given. This part consists of three separate stages. First of all, the proposed method was tested on the CEC’17 benchmark functions, and statistical analyses were made and supported by figures. Then the proposed method is adapted to five real-world engineering problems and the performance analyses of the proposed method and the standard TSO method are examined. Finally, the proposed methods and the TSO method on feature selection were adapted and performance analyses were carried out on classification problems. In the last section, Section 6, the conclusion part is included.

Literature review

Various optimization methods have been proposed in the literature that can be used in optimization problems. These methods have become very popular not only in computer science but also in other research areas (Altay, 2022a). Complex reliability allocation problems (Negi et al., 2021), traveling salesman problem (Mzili, Riffi & Mzili, 2022), association rule mining (Altay & Alatas, 2021), dynamic ship routing and scheduling problem (Das et al., 2022), laser cutting process (Madić et al., 2022), machine learning (Altay & Varol, 2023), and process synthesis problem (Altay, 2022b) are some of them. There are studies comparing the performance of metaheuristic methods (Sadhu et al., 2023).

There is no best optimization algorithm to solve all problems. While an optimization algorithm may solve one problem very well, it may not achieve successful results in another. This encourages researchers to propose new methods and improve existing ones. When the literature is examined, it is seen that many optimization algorithms have been proposed. Some of those include the group teaching optimization algorithm (Zhang & Jin, 2020), dwarf mongoose optimization algorithm (Agushaka, Ezugwu & Abualigah, 2022), chimp optimization algorithm (Khishe & Mosavi, 2020), material generation algorithm (Talatahari, Azizi & Gandomi, 2021), social mimic optimization algorithm (Balochian & Baloochian, 2019), arithmetic trigonometric optimization algorithm (Devan et al., 2022), fertilization optimization algorithm (Devan et al., 2022), African vultures optimization algorithm (Abdollahzadeh, Gharehchopogh & Mirjalili, 2021), aquila optimizer (Abualigah et al., 2021b), circle search algorithm (Qais et al., 2022), the water optimization algorithm (Daliri & Asghari, 2022), and the gold rush optimizer (Zolfi, 2023). The transient search algorithm is one of the physics-based metaheuristic methods that have emerged recently. This method has been tested on some problems, but as far as we know, there is no study related to the development of the method. The fact that it is a new method and that no improvement has been made in this area yet has been our source of motivation.

It is seen that the transient search algorithm has been successfully used in PEM fuel cell modeling (Hasanien et al., 2022), IoT intrusion detection system (Fatani et al., 2021), optimum allocation of more than one distributed generator in the radial electricity distribution network (Bhadoriya & Gupta, 2022), and improving the voltage ride-through capability of the wind turbine (Qais & Hasanien, 2020).

Chaos theory has been extensively employed to enhance exploration and exploitation as nonlinear theory has undergone constant study and improvement. Numerous metaheuristic algorithms’ premature convergence issues have been effectively resolved using chaos theory. Many researchers have added chaotic mapping mechanisms to various metaheuristic algorithms to augment the algorithm’s capacity to find optimum solutions, improve random diversification, and obtain optimal or sub-optimal answers in complicated multi-modal circumstances (Arora & Anand, 2019). The application of chaos theory to different metaheuristic methods can be summarized in Table 1.

Table 1 Literature review on chaos theorem of metaheuristic methods.

Ref.	Year	Proposed model	Chaotic maps	Application	
Farah & Belazi (2018)	2018	Jaya algorithm	2D cross chaotic map	Benchmark function	
Zhang et al. (2018)	2018	Bacterial foraging optimization	Logistic map	Benchmark function	
Sayed, Khoriba & Haggag (2018)	2018	Salp swarm algorithm	Ten different chaotic maps	Benchmark function and feature selection	
Tuba et al. (2018)	2018	Elephant herding optimization	Two different chaotic maps	Benchmark function	
Rizk-Allah, Hassanien & Bhattacharyya (2018)	2018	Crow search algorithm	Ten different chaotic maps	Benchmark function and real-world engineering design problem	
Kaur & Arora (2018)	2018	Whale optimization algorithm	Ten different chaotic maps	Benchmark function	
Sayed, Darwish & Hassanien (2018)	2018	Multi-verse optimization algorithm	Ten different chaotic maps	Real-world engineering design problem	
Arora & Anand (2019)	2018	Grasshopper optimization algorithm	Ten different chaotic maps	Benchmark function	
Li et al. (2019)	2019	Moth-flame optimization	Ten different chaotic maps	Benchmark function and real-world engineering design problem	
Sayed, Tharwat & Hassanien (2019)	2019	Dragonfly algorithm	Ten different chaotic maps	Feature selection	
Demir, Tuncer & Kocamaz (2020)	2020	Chaotic
optimization algorithm	Logistic-sine chaotic map	Benchmark function and real-world engineering design problem	
Bingol & Alatas (2020)	2020	Optics inspired optimization	Five different chaotic maps	Benchmark function and real-world engineering design problem	
Varol Altay & Alatas (2020)	2020	Bird swarm algorithm	Ten different chaotic maps	Benchmark function and real-world engineering design problems	
Pierezan et al. (2021)	2020	Coyote optimization algorithm	Tinkerbell chaotic map	Truss optimization problems	
Gharehchopogh, Maleki & Dizaji (2021)	2021	Vortex search algorithm	Ten different chaotic maps	Feature selection	
Yang et al. (2021)	2021	Spherical evolution algorithm	Twelve different chaotic maps	Benchmark function	
Mohammed & Rashid (2021)	2021	Fitness-dependent optimizer	Ten different chaotic maps	Benchmark function and real-world engineering design problems	
Zhang & Ding (2021)	2021	Sparrow search
algorithm	Logistic map	Benchmark function and stochastic configuration network	
Li et al. (2022)	2021	Arithmetic optimization algorithm	Ten different chaotic maps	Benchmark function and real-world engineering design problems	
Kutlu Onay & Aydemіr (2022)	2021	Hunger games search optimization	Ten different chaotic maps	Benchmark function and real-world engineering design problems	
Altay (2022c)	2022	Slime mould optimization	Ten different chaotic maps	Benchmark function and real-world engineering design problems	
Abualigah & Diabat (2022)	2022	Group search optimizer	Five different chaotic maps	Feature selection	

Tso algorithm

In this section, the background of the TSO algorithm and the operation of the method are discussed. Pseudo code of TSO algorithm is given.

Background of transient search optimization algorithm

The transient performance of electrical circuits has been the inspiration for the metaheuristic optimization method TSO, which has been proposed in recent years. Electrical circuits contain different elements that store energy. These can be capacitors (C), inductors (L), or a combination of both (LC). Generally, an electrical circuit containing a resistor (R), C, or L has a transient response and a steady-state response. This situation is shown in Eq. (1). If the electrical circuit contains an energy storage element together with the resistor, these circuits are classified as first-order circuits. If there are two energy storage elements next to the resistor in the electrical circuit, they are called second-order circuits. The switching of such circuits cannot be changed until the steady-state values of R and L are reached. The transient response of the first-order circuit is calculated by the differential equation in Eq. (2). Equation (2) can be solved to find the solution of x(t) shown in Eq. (3).

(1) Completeresponse=Transientresponse+Finalresponse

(2) ddtx(t)+x(t)τ=K

(3) x(t)=x(∞)+(x(0)−x(∞))e−tτ

where time t, x(t) can be called the capacitor voltage v(t) of the RC circuit or the inductor current i(t) of the RL circuit. τ is called the time constant of the circuit. τ=RC and τ=L/R are for circuit RC and RL, respectively, and K is a constant based on the initial value of x(0). x(∞) is the final response value. The transient response of a quadratic circuit is calculated using the differential equation shown in Eq. (4). The solution of the quadratic differential equation is shown in Eq. (5). Here, the response of the RLC circuit is considered a low-damped response.

(4) d2dt2x(t)+2αddtx(t)+w02x(t)=f(t)

(5) x(t)=e−αt(B1cos⁡((2πfdt))+B2((2πfdt)))+x(∞)

where α is the damping coefficient, w0 is the resonant frequency, fd is the damped resonance frequency, and B1 ve B2 are fixed values. The low damped response occurs when α<w0 causes damped oscillations of the transient response of the RLC circuit.

Transient optimization algorithm

The working logic of the TSO algorithm is similar to the working logic of other metaheuristic optimization algorithms and consists of three steps. In the first of these steps, the initial population is created by creating search agents within the lower and upper limits of the exploration area. The second step is called the exploration phase. In this step, the best solution is sought. The third and final step is called the exploitation stage, and it is aimed at reaching a steady state or the best solution. Search agents in the initial population are randomly generated as in Eq. (6).

(6) Y=lb+rand×(ub−lb)

where the lb value represents the lower limit of the search area, and the ub value the upper limit of the search area. The rand value represents a uniformly distributed random sequence of numbers. The second step, the discovery phase, is designed by inspired by the oscillations of the second order RLC circuits around zero. However, the use of TSO here is inspired by the exponential decay of the first order discharge. The r1 value, which is a random number, is used to balance the exploration ( r1≥0.5) and exploitation (r1<0.5) phases. The use and mathematical model of TSO is shown in Eq. (7), inspired by Eqs. (3) and (5). TSO’s best solution Yl∗, simulates the steady state or final value (x(∞)) of the electrical circuit, also B1=B2=|Yl−C1.Yl∗|.

(7) f(x)={Yl∗+(Yl−C1.Y1∗)e−T,r1<0.5Yl∗+e−T[cos⁡(2πT)+sin(2πT)]|Yl−C1.Yl∗|,r1≥0.5

(8) T=2×z×r2−z

(9) C1=k×z×r3+1

(10) z = 2−2(l/Lmax)

In Eq. (10), z represents a value ranging from 2 to 0 as understood from the equation. The value l represents the number of iterations, T and C1 are random coefficients, r1, r2 and r3 are evenly distributed random numbers and take values between 0 and 1. Yl∗ indicates the best position, k is a constant and Lmax is the maximum number of iterations. The balance between exploration and exploitation processes is achieved with a T coefficient ranging from −2 to 2. Exploitation phase is obtained when T>0 (to the smallest value) and exploration process is obtained when T<0 (to the highest value). The pseudo-code of TSO is given in Algorithm 1.

Algorithm 1 Pseudo-code of TSO algorithm.

Initialize the population and the best positions Yl, Yl∗	
Evaluate the cost function of the population	
while l<Lmax	
  Update the values of T and C1 using Eqs. (8) and (9)	
  do all populations Yl	
    Update the population place by Eq. (7)	
  end do	
  Calculate the cost function of all new population	
  Update the best value if the recent cost function is less than the previous best cost function	
   l=l+1	
end while	
Output the best value Yl∗	

Chaotic maps

In a nonlinear, dynamical system that is non-periodic, non-convergent, and bounded, chaos is a deterministic, random-like technique. Chaos is the randomness of a straightforward deterministic dynamical system in mathematics, and chaotic systems can be thought of as sources of randomness. Although chaos appears random and unpredictable, it also has a certain degree of pattern. Instead of using random variables, chaos uses chaotic variables (Arora & Anand, 2019).

Numbers generated by chaotic maps have been used successfully in a variety of applications. In general, chaotic maps have three basic qualities: ergodicity, beginning conditions, and semi-stochastic properties. Researchers in a variety of domains, including ecology, medicine, economics, and engineering applications, are drawn to chaotic processes. It is also extensively utilized to improve the performance of optimization algorithms (Altay, 2022c).

The effectiveness of swarm intelligence algorithms in tackling a variety of challenging optimization issues has been demonstrated. As a result, further developing these algorithms became a hotly debated scientific subject. The use of chaotic maps in place of random values in swarm intelligence algorithms is one of the more frequently employed innovations. Improved search is anticipated because chaotic maps produce numbers that are non-repeatable and ergodic (Tuba et al., 2018).

Chaotic map applications have proven successful in enhancing the stochastic structure of the optimization techniques in a variety of research. The standard TSO’s global convergence time was accelerated in this work by the introduction of ten separate chaotic maps, which also prevented the standard TSO from being stuck in local solutions. There are many types of chaotic maps that are employed, including the gauss, circle, sine, logistic, piecewise, iterative, singer, tent, and sinusoidal maps. Table 2 presents an explanation of the variables and equations linked to these chaotic maps. The graphics of the ten chaotic maps are shown in Fig. 1.

Table 2 Variables and equations of chaotic maps.

CM no.	CM name	CM equation	
1	Chebyshev map	Xn+1=cos(kcos−1xn)	
2	Circle map	Xn+1=Xn+b−(a2π)sin⁡(2πXn)mod(1) a=0.5 and b=0.2	
3	Gauss map	Xn+1={0,Xn=01Xnmod(1),Xn∈(0,1), 1Xnmod(1)=1Xn−1Xn	
4	Iterative map	Xn+1=sin(aπxn), a=0.7	
5	Logistic map	Xn+1=aXn(1−Xn), a=4	
6	Piecewise map	xn+1={xnP,0≤xn<Pxn−P0.5−P,P≤xn<0.51−P−xn0.5−P,0.5≤xn<1−P1−xnP,1−P≤xn<1, P=0.4	
7	Sine map	Xn+1=a4sin(πxn), 0<a≤4	
8	Singer map	Xn+1=μ(7.86xn−23.31xn2+28.75xn3−13.302875xn4), µ = 1.07	
9	Sinusoidal map	Xn+1=axn2sin(πXn), a=2.3 and X0=0.7	
10	Tent map	f(x)={Xn/0.7,Xn<010/3Xn(1−Xn),otherwise	

Figure 1 Demonstration of chaotic maps.

Chaotic tso algorithm

Metaheuristic optimization algorithms suffer from population diversity and early convergence. In addition, the metaheuristic optimization algorithm needs a balance between exploration and exploitation in order to produce an effective solution. Chaotic maps are one of the most effective ways to increase population diversity and the quality of metaheuristic optimization methods. Thus, both the search sensitivity and convergence speed of the metaheuristic optimization method, which are improved with the chaotic map, are improved. Most metaheuristic optimization methods have an exploration and exploitation phase. The exploration phase is used to search the search area as wide as possible, regardless of whether the method is stuck at the local optimum. In the exploitation phase, the method aims to find the best possible solution in a limited area of the search space by finding the most promising area. While the method spends too much time on the exploitation process, which only leads to a local search, spending too much time on the exploration process results in a random search.

Although the newly proposed TSO algorithm has been successfully applied in different applications in the literature, it has some disadvantages, as we mentioned above. Therefore, in order to improve the search space of the TSO algorithm, chaotic maps, which are accepted in the literature, have been added to the dynamic behavior and optimization algorithms to make the search space stronger. Equation (7) is used for exploration and exploitation in the TSO algorithm, whose working principle is very clear. Random variables affecting Eq. (7) are regular random variable values r2 and r3 in Eqs. (8) and (9), respectively. In this study, instead of the r3 value, which is the stochastic component of the TSO algorithm in Eq. (9), chaotic maps with different mathematical equations listed in Table 2 were applied. The mathematical equation is shown in the equation below.

(11) C1=k×z×CMV+1

Here, CMV, that is, sequentially generated chaotic map value is used instead of r3 value. Proposed algorithms using 10 different chaotic map; chebyshev map (CTSO-1), circle map (CTSO-2), gauss map (CTSO-3), iterative map (CTSO-4), logistic map (CTSO-5), piecewise map (CTSO-6), sine map (CTSO-7), singer map (CTSO-8), sinusoidal map (CTSO-9) and tent map (CTSO-10). The flowchart of the CTSO is shown in Fig. 2.

Figure 2 Flowchart of CTSOs.

Also, the complexity of the proposed CTSO algorithms can be expressed using big-oh notation. The process of the CTSO algorithm starts with the random generation of search agents in the first step, evaluates the search agents using the cost function in the second step, and updates the search agents to the function evaluation value in the third step. Here, the first step is denoted by O(N), where N is the number of search agents. In the second step, the search agents enter the while loop, which has the maximum iteration ( Lmax). The complexity of function evaluations of all search agents is expressed as O(N∗Lmax). And finally, in the third step, the complexity of updating all search agents with a size (D) for total iterations is expressed as O(N∗Lmax∗D).

Results and discussion

In order to evaluate the performance of the methods proposed in the study, CEC’17 test functions, five different real-world problems, and feature selection problems consisting of 10 different real-world datasets were applied. The results obtained were compared in detail under this section and the performance analyses of the methods were carried out. In all tables, the use of bold demonstrates the best result attained. TSO and CTSOs are taken as constant parameters with k value of 1 and z value of [0,2]. The experimental tests are performed using MATLAB R2021a and the whole test is executed on a PC (Intel (R) Core (TM) i9–10900k CPU @ 3.70 GHz (10 CPUs), 32 GB, Windows 10–64 bits).

Benchmark function

The performance evaluation of ten distinct CTSO and TSO methods was conducted using the IEEE Congress on Evolutionary Computation (CEC) test functions. The CEC’17 test suite comprises a total of 29 test functions, encompassing a diverse range of function types such as unimodal, multimodal, hybrid, and composition functions. The algorithm’s convergence performance is assessed using unimodal functions, namely f1 and f3, while the presence of early convergence and local fixation issues is evaluated using multimodal functions, specifically f4 through f10. The assessment of the capacity to evade local optima, which are characterized by numerous local optima, and the equilibrium between exploration and exploitation is carried out through the utilization of hybrid and composition functions (f11–f20 and f21–f30).

The lower and upper bounds of all functions included in the CEC’17 test suite are between −100 and 100. In order to make a fair evaluation under equal conditions, the number of evaluations was chosen as 1,000 and the population as 30. Algorithms were run 30 times in all experiments, and the results of mean (AVG), standard deviation (STD), minimum (MIN), and Friedman mean rank (MR) values are presented in Tables 3A–3C in a comparative manner. According to the MR value, CTSO-9 and CTSO-5 showed the best performance in unimodal benchmark functions. Of the multimodal functions, CTSO-3 showed the best performance in 3, CTSO-9 in 2, CTSO-3 and CTSO-9 in 1, and CTSO-5 in 1 of them. Of the hybrid and composition functions, CTSO-3 showed the best performance in 11, CTSO-9 in 6, CTSO-3 and CTSO-9 in 2, and CTSO-10 in 1 of them. In Table 4, the average Friedman mean rank values based on the MR values of all benchmark functions are given. When Table 4 is examined according to the statistical analysis results, CTSO-3 gives the best performance, followed by CTSO-9 with a close value. CTSO-8 performed worse than the original TSO. The convergence performance of the algorithms on the CEC’17 test functions is also given in Fig. 3 according to the best values of the algorithms. Figure 4 presents a boxplot of CEC’17 test functions.

Table 3 Experimental results on CEC’17 test functions.

Table 3-A	
Algorithm	AVG	STD	MIN	MR	Algorithm	AVG	STD	MIN	MR	
f1	f3	
TSO	4.95E+10	8.21E+09	2.48E+10	7.20	TSO	9.11E+04	3.47E+03	8.10E+04	5.47	
CTSO-1	4.92E+10	8.39E+09	3.13E+10	7.13	CTSO-1	9.14E+04	3.08E+03	8.21E+04	5.13	
CTSO-2	4.99E+10	8.71E+09	3.24E+10	7.33	CTSO-2	9.09E+04	3.33E+03	8.32E+04	5.10	
CTSO-3	1.57E+08	1.45E+08	1.86E+07	1.53	CTSO-3	1.55E+05	6.93E+04	6.08E+04	9.30	
CTSO-4	4.79E+10	6.77E+09	3.44E+10	6.33	CTSO-4	9.13E+04	5.09E+03	7.18E+04	6.27	
CTSO-5	4.97E+10	7.32E+09	3.54E+10	6.87	CTSO-5	9.02E+04	5.42E+03	6.97E+04	4.77	
CTSO-6	4.96E+10	7.20E+09	3.52E+10	6.90	CTSO-6	9.19E+04	2.38E+03	8.70E+04	5.53	
CTSO-7	4.98E+10	9.49E+09	2.63E+10	7.20	CTSO-7	9.06E+04	4.41E+03	7.63E+04	5.03	
CTSO-8	5.01E+10	9.07E+09	3.54E+10	6.80	CTSO-8	9.04E+04	6.25E+03	6.91E+04	5.53	
CTSO-9	1.83E+08	2.95E+08	3.75E+07	1.47	CTSO-9	1.36E+05	4.90E+04	6.55E+04	8.90	
CTSO-10	4.95E+10	8.31E+09	3.34E+10	7.23	CTSO-10	9.13E+04	3.25E+03	8.17E+04	4.97	
f4	f5	
TSO	1.23E+04	3.21E+03	4.76E+03	7.17	TSO	9.12E+02	3.32E+01	8.37E+02	6.57	
CTSO-1	1.19E+04	2.29E+03	7.99E+03	6.67	CTSO-1	9.19E+02	3.13E+01	8.51E+02	7.77	
CTSO-2	1.20E+04	2.21E+03	7.98E+03	6.83	CTSO-2	9.06E+02	4.49E+01	8.16E+02	6.30	
CTSO-3	6.16E+02	6.91E+01	5.26E+02	1.50	CTSO-3	7.94E+02	4.53E+01	7.10E+02	1.77	
CTSO-4	1.25E+04	2.95E+03	8.86E+03	7.30	CTSO-4	9.16E+02	3.51E+01	8.30E+02	7.23	
CTSO-5	1.13E+04	3.14E+03	5.83E+03	6.20	CTSO-5	9.09E+02	3.83E+01	7.97E+02	6.77	
CTSO-6	1.23E+04	2.99E+03	7.30E+03	7.47	CTSO-6	9.17E+02	3.78E+01	8.13E+02	7.63	
CTSO-7	1.24E+04	3.15E+03	7.15E+03	7.13	CTSO-7	9.07E+02	3.14E+01	8.41E+02	6.23	
CTSO-8	1.31E+04	2.72E+03	8.01E+03	7.83	CTSO-8	9.20E+02	2.67E+01	8.76E+02	7.43	
CTSO-9	6.07E+02	4.59E+01	4.94E+02	1.50	CTSO-9	7.83E+02	5.52E+01	6.87E+02	1.73	
CTSO-10	1.17E+04	3.39E+03	6.65E+03	6.40	CTSO-10	9.10E+02	2.71E+01	8.34E+02	6.57	
f6	f7	
TSO	6.85E+02	8.78E+00	6.59E+02	7.23	TSO	1.44E+03	5.86E+01	1.31E+03	6.73	
CTSO-1	6.83E+02	8.22E+00	6.64E+02	6.17	CTSO-1	1.43E+03	5.71E+01	1.31E+03	5.97	
CTSO-2	6.82E+02	7.18E+00	6.67E+02	5.97	CTSO-2	1.43E+03	6.92E+01	1.24E+03	6.03	
CTSO-3	6.67E+02	1.08E+01	6.47E+02	2.17	CTSO-3	1.46E+03	1.27E+02	1.25E+03	6.70	
CTSO-4	6.83E+02	7.81E+00	6.66E+02	6.07	CTSO-4	1.43E+03	5.07E+01	1.26E+03	5.87	
CTSO-5	6.86E+02	8.16E+00	6.64E+02	7.17	CTSO-5	1.41E+03	7.34E+01	1.25E+03	4.70	
CTSO-6	6.84E+02	1.02E+01	6.64E+02	6.37	CTSO-6	1.43E+03	5.70E+01	1.30E+03	6.23	
CTSO-7	6.87E+02	8.70E+00	6.61E+02	7.73	CTSO-7	1.42E+03	6.65E+01	1.22E+03	5.30	
CTSO-8	6.86E+02	6.16E+00	6.75E+02	7.20	CTSO-8	1.43E+03	5.03E+01	1.31E+03	5.80	
CTSO-9	6.70E+02	1.04E+01	6.52E+02	2.73	CTSO-9	1.44E+03	1.36E+02	1.18E+03	6.10	
CTSO-10	6.86E+02	7.40E+00	6.69E+02	7.20	CTSO-10	1.45E+03	5.05E+01	1.30E+03	6.57	
f8	f9	
TSO	1.12E+03	2.52E+01	1.08E+03	6.63	TSO	1.05E+04	1.34E+03	8.17E+03	7.13	
CTSO-1	1.12E+03	3.33E+01	1.04E+03	6.50	CTSO-1	1.03E+04	1.32E+03	7.94E+03	6.70	
CTSO-2	1.13E+03	2.10E+01	1.09E+03	7.93	CTSO-2	1.04E+04	1.07E+03	8.43E+03	6.87	
CTSO-3	1.01E+03	4.72E+01	9.17E+02	1.73	CTSO-3	7.25E+03	1.57E+03	4.58E+03	2.00	
CTSO-4	1.12E+03	2.95E+01	1.07E+03	6.43	CTSO-4	1.01E+04	1.31E+03	6.02E+03	6.30	
CTSO-5	1.11E+03	3.19E+01	1.02E+03	5.63	CTSO-5	1.03E+04	1.23E+03	7.82E+03	7.07	
CTSO-6	1.13E+03	2.50E+01	1.07E+03	7.37	CTSO-6	1.03E+04	1.19E+03	7.59E+03	7.10	
CTSO-7	1.13E+03	2.54E+01	1.05E+03	6.97	CTSO-7	1.05E+04	1.44E+03	7.58E+03	6.70	
CTSO-8	1.13E+03	2.69E+01	1.08E+03	7.73	CTSO-8	1.08E+04	1.46E+03	7.90E+03	7.73	
CTSO-9	1.01E+03	4.35E+01	9.09E+02	1.50	CTSO-9	7.43E+03	1.65E+03	4.89E+03	2.40	
CTSO-10	1.13E+03	1.82E+01	1.10E+03	7.57	CTSO-10	9.85E+03	1.47E+03	7.05E+03	6.00	
f10	f11	
TSO	8.69E+03	5.78E+02	7.64E+03	5.73	TSO	9.85E+03	3.09E+03	4.06E+03	6.40	
CTSO-1	8.95E+03	6.98E+02	7.58E+03	6.83	CTSO-1	1.05E+04	2.80E+03	5.00E+03	7.07	
CTSO-2	8.83E+03	8.47E+02	7.30E+03	6.77	CTSO-2	9.79E+03	2.15E+03	6.53E+03	6.77	
CTSO-3	6.73E+03	1.33E+03	4.70E+03	2.37	CTSO-3	1.68E+03	3.27E+02	1.24E+03	1.40	
CTSO-4	8.86E+03	5.27E+02	8.05E+03	6.43	CTSO-4	1.04E+04	2.00E+03	6.26E+03	7.10	
CTSO-5	9.01E+03	6.78E+02	7.64E+03	7.40	CTSO-5	1.01E+04	2.33E+03	3.00E+03	7.20	
CTSO-6	8.99E+03	7.03E+02	7.82E+03	7.33	CTSO-6	9.57E+03	2.20E+03	4.63E+03	6.60	
CTSO-7	8.49E+03	5.62E+02	7.30E+03	5.40	CTSO-7	1.01E+04	2.73E+03	4.57E+03	6.90	
CTSO-8	8.93E+03	7.24E+02	7.33E+03	7.27	CTSO-8	1.04E+04	2.16E+03	5.38E+03	7.50	
CTSO-9	7.66E+03	1.41E+03	4.85E+03	4.03	CTSO-9	2.52E+03	2.84E+03	1.33E+03	1.90	
CTSO-10	8.81E+03	6.64E+02	7.56E+03	6.43	CTSO-10	1.01E+04	1.77E+03	7.43E+03	7.17	
Table 3-B	
Algorithm	AVG	STD	MIN	MR	Algorithm	AVG	STD	MIN	MR	
f12	f13	
TSO	1.01E+10	4.02E+09	3.60E+09	7.70	TSO	2.70E+09	3.49E+09	2.27E+08	6.57	
CTSO-1	7.78E+09	3.34E+09	2.90E+09	6.20	CTSO-1	2.53E+09	3.16E+09	4.33E+08	7.50	
CTSO-2	8.51E+09	3.52E+09	2.90E+09	6.70	CTSO-2	2.84E+09	4.16E+09	6.69E+07	7.00	
CTSO-3	2.04E+07	2.31E+07	6.18E+05	1.53	CTSO-3	5.68E+04	4.66E+04	1.14E+04	1.33	
CTSO-4	8.93E+09	3.95E+09	2.87E+09	6.90	CTSO-4	2.46E+09	3.43E+09	8.13E+07	6.57	
CTSO-5	8.59E+09	3.59E+09	3.45E+09	6.57	CTSO-5	2.92E+09	3.56E+09	7.57E+08	7.53	
CTSO-6	9.35E+09	3.31E+09	3.23E+09	7.30	CTSO-6	1.89E+09	2.52E+09	1.55E+08	6.40	
CTSO-7	9.48E+09	4.68E+09	3.75E+09	7.20	CTSO-7	2.27E+09	2.67E+09	1.58E+08	6.70	
CTSO-8	9.97E+09	4.17E+09	3.87E+09	7.43	CTSO-8	2.98E+09	4.32E+09	1.60E+08	7.20	
CTSO-9	2.47E+07	3.62E+07	1.01E+06	1.47	CTSO-9	4.67E+07	2.36E+08	1.40E+04	1.80	
CTSO-10	9.12E+09	3.70E+09	3.15E+09	7.00	CTSO-10	3.02E+09	3.45E+09	2.00E+08	7.40	
f14	f15	
TSO	7.27E+06	5.82E+06	3.40E+05	7.27	TSO	4.34E+08	6.96E+08	7.91E+06	6.07	
CTSO-1	1.01E+07	2.04E+07	1.53E+05	6.33	CTSO-1	5.47E+08	5.45E+08	1.82E+06	7.47	
CTSO-2	7.56E+06	1.02E+07	2.16E+05	7.27	CTSO-2	4.20E+08	4.48E+08	2.70E+07	6.97	
CTSO-3	1.45E+06	3.77E+06	3.86E+03	2.43	CTSO-3	1.64E+04	1.28E+04	3.64E+03	1.50	
CTSO-4	5.56E+06	5.02E+06	4.65E+05	6.57	CTSO-4	4.58E+08	5.59E+08	1.23E+07	7.20	
CTSO-5	6.28E+06	5.64E+06	2.45E+05	6.73	CTSO-5	5.61E+08	7.55E+08	6.00E+06	7.17	
CTSO-6	8.11E+06	7.55E+06	2.68E+05	7.50	CTSO-6	3.62E+08	5.47E+08	1.49E+07	6.33	
CTSO-7	4.99E+06	3.97E+06	3.18E+05	6.07	CTSO-7	4.26E+08	3.91E+08	1.32E+07	7.40	
CTSO-8	7.43E+06	7.47E+06	8.20E+04	6.57	CTSO-8	5.19E+08	5.05E+08	2.04E+07	7.80	
CTSO-9	9.26E+05	1.87E+06	5.91E+03	2.33	CTSO-9	1.69E+04	1.30E+04	2.28E+03	1.50	
CTSO-10	1.21E+07	3.24E+07	2.11E+05	6.93	CTSO-10	3.68E+08	4.55E+08	4.16E+06	6.60	
f16	f17	
TSO	5.48E+03	9.37E+02	8.03E+03	7.17	TSO	3.45E+03	7.68E+02	2.30E+03	7.00	
CTSO-1	5.74E+03	1.07E+03	8.56E+03	7.43	CTSO-1	4.19E+03	4.53E+03	2.48E+03	6.43	
CTSO-2	5.17E+03	9.01E+02	7.32E+03	6.30	CTSO-2	3.50E+03	1.28E+03	2.64E+03	5.47	
CTSO-3	3.55E+03	7.81E+02	5.57E+03	2.07	CTSO-3	2.74E+03	3.60E+02	2.14E+03	3.00	
CTSO-4	5.58E+03	1.08E+03	7.68E+03	6.97	CTSO-4	4.32E+03	3.29E+03	2.02E+03	6.93	
CTSO-5	5.16E+03	6.95E+02	6.30E+03	6.37	CTSO-5	3.35E+03	8.24E+02	2.63E+03	6.23	
CTSO-6	5.53E+03	9.81E+02	8.44E+03	7.13	CTSO-6	3.56E+03	1.03E+03	2.42E+03	6.20	
CTSO-7	5.71E+03	1.05E+03	7.73E+03	7.37	CTSO-7	5.03E+03	6.07E+03	2.42E+03	6.20	
CTSO-8	5.44E+03	9.72E+02	8.31E+03	6.57	CTSO-8	4.64E+03	4.14E+03	2.47E+03	7.37	
CTSO-9	3.49E+03	6.43E+02	5.33E+03	1.93	CTSO-9	2.94E+03	3.69E+02	2.29E+03	4.53	
CTSO-10	5.29E+03	1.03E+03	7.54E+03	6.70	CTSO-10	4.58E+03	4.16E+03	2.58E+03	6.63	
f18	f19	
TSO	9.48E+07	1.03E+08	5.85E+06	7.57	TSO	4.94E+08	4.74E+08	1.90E+07	6.77	
CTSO-1	6.32E+07	6.84E+07	1.73E+06	6.67	CTSO-1	3.97E+08	3.36E+08	2.65E+07	6.60	
CTSO-2	7.71E+07	9.13E+07	1.58E+06	6.90	CTSO-2	3.78E+08	3.98E+08	2.84E+07	5.97	
CTSO-3	1.20E+06	3.65E+06	9.33E+04	1.53	CTSO-3	1.68E+05	5.67E+05	2.92E+03	1.43	
CTSO-4	1.08E+08	1.26E+08	5.89E+06	7.73	CTSO-4	5.81E+08	4.94E+08	8.05E+07	7.23	
CTSO-5	6.37E+07	7.02E+07	1.02E+06	6.30	CTSO-5	5.74E+08	4.75E+08	2.91E+07	7.37	
CTSO-6	1.36E+08	2.52E+08	2.39E+06	7.60	CTSO-6	5.88E+08	5.07E+08	2.55E+07	7.10	
CTSO-7	5.56E+07	6.10E+07	2.13E+06	6.27	CTSO-7	5.15E+08	4.67E+08	1.45E+07	6.93	
CTSO-8	9.00E+07	1.17E+08	1.06E+06	7.27	CTSO-8	5.74E+08	4.58E+08	3.09E+07	7.33	
CTSO-9	2.68E+06	6.58E+06	3.96E+04	1.87	CTSO-9	6.94E+04	1.61E+05	2.21E+03	1.57	
CTSO-10	5.50E+07	5.24E+07	9.39E+05	6.30	CTSO-10	6.64E+08	6.30E+08	2.52E+07	7.70	
f20	f21	
TSO	2.97E+03	1.91E+02	2.56E+03	5.43	TSO	2.72E+03	4.50E+01	2.63E+03	7.07	
CTSO-1	2.97E+03	1.91E+02	2.52E+03	6.50	CTSO-1	2.72E+03	5.31E+01	2.62E+03	7.20	
CTSO-2	2.97E+03	1.59E+02	2.60E+03	5.80	CTSO-2	2.71E+03	4.52E+01	2.62E+03	6.80	
CTSO-3	2.99E+03	2.50E+02	2.40E+03	6.23	CTSO-3	2.60E+03	5.54E+01	2.49E+03	2.13	
CTSO-4	3.03E+03	2.44E+02	2.53E+03	6.53	CTSO-4	2.71E+03	5.90E+01	2.63E+03	6.43	
CTSO-5	3.01E+03	1.41E+02	2.74E+03	6.60	CTSO-5	2.71E+03	5.16E+01	2.60E+03	6.20	
CTSO-6	2.96E+03	1.78E+02	2.59E+03	5.73	CTSO-6	2.74E+03	4.94E+01	2.64E+03	7.83	
CTSO-7	2.96E+03	1.84E+02	2.66E+03	5.53	CTSO-7	2.72E+03	6.25E+01	2.61E+03	6.97	
CTSO-8	2.99E+03	1.60E+02	2.65E+03	6.17	CTSO-8	2.71E+03	6.74E+01	2.60E+03	5.93	
CTSO-9	3.06E+03	2.49E+02	2.56E+03	6.67	CTSO-9	2.60E+03	5.86E+01	2.51E+03	2.30	
CTSO-10	2.92E+03	2.16E+02	2.54E+03	4.80	CTSO-10	2.72E+03	4.95E+01	2.61E+03	7.13	
Table 3-C	
Algorithm	AVG	STD	MIN	MR	Algorithm	AVG	STD	MIN	MR	
f22	f23	
TSO	9.47E+03	9.40E+02	6.59E+03	6.33	TSO	3.46E+03	1.44E+02	3.19E+03	5.80	
CTSO-1	9.66E+03	8.69E+02	6.58E+03	7.13	CTSO-1	3.54E+03	1.60E+02	3.29E+03	7.43	
CTSO-2	9.86E+03	7.73E+02	7.16E+03	7.73	CTSO-2	3.49E+03	1.85E+02	3.22E+03	6.07	
CTSO-3	8.50E+03	1.45E+03	5.29E+03	3.93	CTSO-3	3.36E+03	1.62E+02	3.01E+03	3.83	
CTSO-4	9.36E+03	9.72E+02	6.94E+03	6.20	CTSO-4	3.49E+03	1.93E+02	3.05E+03	6.20	
CTSO-5	9.34E+03	9.78E+02	6.16E+03	5.77	CTSO-5	3.47E+03	1.95E+02	3.16E+03	5.47	
CTSO-6	9.52E+03	8.58E+02	7.13E+03	6.57	CTSO-6	3.53E+03	1.80E+02	3.28E+03	6.87	
CTSO-7	9.56E+03	6.84E+02	7.67E+03	6.57	CTSO-7	3.52E+03	1.73E+02	3.20E+03	7.03	
CTSO-8	9.50E+03	1.12E+03	6.35E+03	6.47	CTSO-8	3.55E+03	2.67E+02	3.21E+03	6.77	
CTSO-9	8.06E+03	1.86E+03	2.84E+03	3.03	CTSO-9	3.36E+03	1.61E+02	2.98E+03	4.10	
CTSO-10	9.58E+03	5.84E+02	8.57E+03	6.27	CTSO-10	3.48E+03	1.47E+02	3.18E+03	6.43	
f24	f25	
TSO	3.71E+03	2.08E+02	3.37E+03	6.80	TSO	4.68E+03	4.06E+02	3.79E+03	8.03	
CTSO-1	3.63E+03	1.90E+02	3.31E+03	5.30	CTSO-1	4.48E+03	4.97E+02	3.56E+03	6.23	
CTSO-2	3.75E+03	2.98E+02	3.36E+03	6.70	CTSO-2	4.50E+03	3.61E+02	3.72E+03	6.60	
CTSO-3	3.53E+03	2.47E+02	3.04E+03	4.33	CTSO-3	3.02E+03	4.51E+01	2.93E+03	1.57	
CTSO-4	3.72E+03	2.39E+02	3.38E+03	6.40	CTSO-4	4.58E+03	4.62E+02	3.87E+03	7.27	
CTSO-5	3.76E+03	2.26E+02	3.31E+03	7.17	CTSO-5	4.59E+03	3.97E+02	3.82E+03	7.50	
CTSO-6	3.74E+03	2.66E+02	3.35E+03	6.60	CTSO-6	4.65E+03	4.67E+02	3.79E+03	7.43	
CTSO-7	3.72E+03	3.32E+02	3.38E+03	5.93	CTSO-7	4.54E+03	4.32E+02	3.64E+03	6.70	
CTSO-8	3.76E+03	3.23E+02	3.42E+03	6.50	CTSO-8	4.54E+03	3.53E+02	4.00E+03	7.17	
CTSO-9	3.50E+03	1.88E+02	3.21E+03	3.43	CTSO-9	3.01E+03	3.92E+01	2.93E+03	1.43	
CTSO-10	3.74E+03	2.92E+02	3.35E+03	6.83	CTSO-10	4.45E+03	3.90E+02	3.79E+03	6.07	
f26	f27	
TSO	1.13E+04	1.15E+03	8.76E+03	7.67	TSO	4.26E+03	4.63E+02	3.57E+03	7.30	
CTSO-1	1.10E+04	1.06E+03	8.23E+03	6.90	CTSO-1	4.21E+03	4.56E+02	3.49E+03	6.40	
CTSO-2	1.13E+04	1.02E+03	9.57E+03	7.73	CTSO-2	4.12E+03	3.83E+02	3.56E+03	6.23	
CTSO-3	8.45E+03	1.44E+03	4.32E+03	1.90	CTSO-3	3.55E+03	2.19E+02	3.23E+03	1.93	
CTSO-4	1.13E+04	1.18E+03	9.20E+03	7.50	CTSO-4	4.16E+03	3.06E+02	3.45E+03	6.93	
CTSO-5	1.07E+04	1.22E+03	8.24E+03	6.00	CTSO-5	4.18E+03	3.88E+02	3.45E+03	7.20	
CTSO-6	1.06E+04	9.86E+02	9.22E+03	5.80	CTSO-6	4.13E+03	3.67E+02	3.64E+03	6.50	
CTSO-7	1.13E+04	1.34E+03	8.39E+03	7.60	CTSO-7	4.09E+03	4.01E+02	3.58E+03	5.97	
CTSO-8	1.06E+04	1.25E+03	8.48E+03	5.63	CTSO-8	4.26E+03	3.71E+02	3.55E+03	7.73	
CTSO-9	9.01E+03	1.50E+03	3.63E+03	2.77	CTSO-9	3.66E+03	2.77E+02	3.28E+03	2.93	
CTSO-10	1.10E+04	1.37E+03	8.98E+03	6.50	CTSO-10	4.23E+03	4.74E+02	3.61E+03	6.87	
f28	f29	
TSO	6.95E+03	7.63E+02	5.11E+03	7.80	TSO	7.09E+03	1.32E+03	5.34E+03	7.30	
CTSO-1	6.78E+03	8.33E+02	5.49E+03	7.23	CTSO-1	6.60E+03	1.07E+03	4.83E+03	5.93	
CTSO-2	6.52E+03	6.75E+02	5.35E+03	6.03	CTSO-2	6.82E+03	1.39E+03	4.71E+03	6.57	
CTSO-3	3.36E+03	4.68E+01	3.28E+03	1.43	CTSO-3	5.59E+03	6.93E+02	4.80E+03	3.00	
CTSO-4	6.59E+03	8.56E+02	5.32E+03	6.63	CTSO-4	6.64E+03	1.65E+03	4.67E+03	5.53	
CTSO-5	6.71E+03	8.21E+02	5.50E+03	6.83	CTSO-5	7.58E+03	3.68E+03	5.25E+03	6.73	
CTSO-6	6.59E+03	9.89E+02	4.39E+03	6.80	CTSO-6	6.87E+03	1.25E+03	5.42E+03	6.70	
CTSO-7	6.78E+03	8.11E+02	5.23E+03	7.17	CTSO-7	6.62E+03	1.13E+03	5.10E+03	6.27	
CTSO-8	6.74E+03	8.69E+02	5.12E+03	6.93	CTSO-8	7.11E+03	1.40E+03	5.01E+03	7.27	
CTSO-9	3.38E+03	8.88E+01	3.26E+03	1.57	CTSO-9	5.73E+03	8.07E+02	4.46E+03	3.53	
CTSO-10	6.90E+03	6.42E+02	5.73E+03	7.57	CTSO-10	7.06E+03	1.55E+03	4.58E+03	7.17	
f30						
Algorithm	AVG	STD	MIN	MR						
TSO	8.51E+08	9.03E+08	1.91E+08	7.43						
CTSO-1	6.09E+08	5.87E+08	7.08E+07	6.37						
CTSO-2	5.23E+08	5.57E+08	8.79E+07	5.87						
CTSO-3	1.87E+06	2.13E+06	5.87E+04	1.50						
CTSO-4	6.15E+08	5.41E+08	3.57E+07	6.57						
CTSO-5	7.89E+08	5.65E+08	4.65E+07	7.57						
CTSO-6	8.63E+08	7.59E+08	1.21E+08	7.20						
CTSO-7	8.42E+08	7.37E+08	8.26E+07	7.43						
CTSO-8	9.36E+08	7.41E+08	1.48E+08	7.93						
CTSO-9	3.02E+06	6.92E+06	3.88E+04	1.50						
CTSO-10	6.63E+08	5.55E+08	8.49E+07	6.63						
Note:

The use of bold demonstrates the best results attained.

Table 4 Experimental results on CEC’17 test functions.

TSO	CTSO-1	CTSO-2	CTSO-3	CTSO-4	CTSO-5	CTSO-6	CTSO-7	CTSO-8	CTSO-9	CTSO-10	
6.87	6.66	6.57	2.66	6.68	6.59	6.83	6.62	7.00	2.85	6.68	

Figure 3 Convergence curve graph of CEC’17 test functions.

Figure 4 Boxplot of CEC’17 test functions.

The results of the fitness functions obtained using the Wilcoxon rank-sum test are given in Table 5. The results obtained were obtained at a significance level of five percent. The results obtained must be less than 0.05 to show a significant advantage. The △, ▽, and ≈ signs in the table indicate the superiority of one of the proposed CTSOs, the TSO is superior, and there is no significant difference between the methods, respectively. When the Table 5 is examined, it is seen that CTSO-3 and CTSO-9 show the best performances by providing superiority to TSO in 26 functions. It can be said that Gaussian and sinusoidal chaotic maps are superior to other chaotic maps and standard TSO in benchmark functions.

Table 5 Wilcoxon rank-sum test on CEC’17 test functions.

	CTSO-1	CTSO-2	CTSO-3	CTSO-4	CTSO-5	CTSO-6	CTSO-7	CTSO-8	CTSO-9	CTSO-10	
f1	8.53E−01	8.65E−01	3.02E−11	1.91E−01	5.89E−01	7.96E−01	9.71E−01	7.28E−01	3.02E−11	7.06E−01	
	≈	≈	Δ	≈	≈	≈	≈	≈	Δ	≈	
f3	9.82E−01	5.30E−01	1.86E−06	2.97E−01	6.00E−01	4.83E−01	6.20E−01	5.20E−01	4.74E−06	9.00E−01	
	≈	≈	∇	≈	≈	≈	≈	≈	∇	≈	
f4	5.69E−01	6.84E−01	3.02E−11	9.00E−01	1.62E−01	9.47E−01	1.00E+00	3.40E−01	3.02E−11	3.04E−01	
	≈	≈	Δ	≈	≈	≈	≈	≈	Δ	≈	
f5	4.55E−01	6.00E−01	1.96E−10	4.92E−01	9.23E−01	2.64E−01	1.81E−01	4.83E−01	6.72E−10	6.00E−01	
	≈	≈	Δ	≈	≈	≈	≈	≈	Δ	≈	
f6	3.48E−01	9.93E−02	6.53E−08	1.41E−01	9.71E−01	3.04E−01	4.29E−01	9.00E−01	1.39E−06	8.19E−01	
	≈	≈	Δ	≈	≈	≈	≈	≈	Δ	≈	
f7	2.64E−01	5.40E−01	5.01E−01	1.09E−01	3.03E−02	6.10E−01	7.01E−02	2.97E−01	3.40E−01	8.53E−01	
	≈	≈	≈	≈	Δ	≈	Δ	≈	≈	≈	
f8	5.30E−01	1.15E−01	2.37E−10	5.20E−01	5.55E−02	5.59E−01	5.40E−01	3.95E−01	3.34E−11	1.81E−01	
	≈	≈	Δ	≈	≈	≈	≈	≈	Δ	≈	
f9	8.07E−01	8.88E−01	2.67E−09	6.73E−01	6.31E−01	8.30E−01	1.00E+00	3.33E−01	3.65E−08	1.76E−01	
	≈	≈	Δ	≈	≈	≈	≈	≈	Δ	≈	
f10	2.28E−01	5.30E−01	4.44E−07	3.33E−01	5.37E−02	1.12E−01	2.71E−01	1.54E−01	6.67E−03	5.30E−01	
	≈	≈	Δ	≈	≈	≈	≈	≈	Δ		
f11	2.90E−01	9.23E−01	3.02E−11	2.12E−01	3.11E−01	9.94E−01	6.20E−01	2.71E−01	8.10E−10	4.73E−01	
	≈	≈	Δ	≈	≈	≈	≈	≈	Δ	≈	
f12	2.07E−02	1.37E−01	3.02E−11	2.40E−01	1.12E−01	5.30E−01	5.20E−01	9.23E−01	3.02E−11	3.79E−01	
	Δ	≈	Δ	≈	≈	≈	≈	≈	Δ	≈	
f13	7.28E−01	8.30E−01	3.02E−11	5.11E−01	4.12E−01	1.91E−01	5.59E−01	8.65E−01	1.21E−10	5.01E−01	
	≈	≈	Δ	≈	≈	≈	≈	≈	Δ	≈	
f14	3.71E−01	6.95E−01	1.60E−07	2.71E−01	6.00E−01	9.23E−01	1.67E−01	7.06E−01	2.02E−08	6.84E−01	
	≈	≈	Δ	≈	≈	≈	≈	≈	Δ	≈	
f15	6.35E−02	1.19E−01	3.02E−11	1.41E−01	6.57E−02	5.49E−01	1.22E−01	3.15E−02	3.02E−11	3.87E−01	
	≈	≈	Δ	≈	≈	≈	≈	Δ	Δ	≈	
f16	3.71E−01	2.97E−01	3.20E−09	8.30E−01	2.40E−01	9.82E−01	4.38E−01	7.06E−01	6.72E−10	3.48E−01	
	≈	≈	Δ	≈	≈	≈	≈	≈	Δ	≈	
f17	5.59E−01	7.01E−02	2.15E−06	6.31E−01	3.79E−01	8.65E−01	6.10E−01	3.79E−01	1.17E−03	7.28E−01	
	≈	≈	Δ	≈	≈	≈	≈	≈	Δ	≈	
f18	1.71E−01	5.40E−01	5.49E−11	7.73E−01	1.19E−01	9.59E−01	6.79E−02	6.10E−01	1.33E−10	1.22E−01	
	≈	≈	Δ	≈	≈	≈	≈	≈	Δ	≈	
f19	7.28E−01	2.64E−01	3.02E−11	4.20E−01	5.59E−01	4.46E−01	9.94E−01	4.92E−01	3.02E−11	3.79E−01	
	≈	≈	Δ	≈	≈	≈	≈	≈	Δ	≈	
f20	4.83E−01	7.17E−01	3.04E−01	3.33E−01	1.58E−01	6.20E−01	9.00E−01	5.01E−01	9.05E−02	5.89E−01	
	≈	≈	≈	≈	≈	≈	≈	≈	≈	≈	
f21	9.94E−01	5.11E−01	3.82E−09	4.29E−01	3.11E−01	1.58E−01	7.39E−01	2.46E−01	3.20E−09	9.59E−01	
	≈	≈	Δ	≈	≈	≈	≈	≈	Δ	≈	
f22	3.71E−01	6.35E−02	2.38E−03	6.84E−01	5.79E−01	7.73E−01	8.88E−01	6.31E−01	3.77E−04	9.23E−01	
	≈	≈	Δ	≈	≈	≈	≈	≈	Δ	≈	
f23	4.84E−02	6.95E−01	1.44E−02	7.62E−01	8.53E−01	1.12E−01	1.58E−01	2.58E−01	1.63E−02	5.30E−01	
	Δ	≈	Δ	≈	≈	≈	≈	≈	Δ	≈	
f24	1.49E−01	7.84E−01	4.23E−03	7.73E−01	2.90E−01	6.10E−01	3.18E−01	8.65E−01	5.97E−05	9.82E−01	
	≈	≈	Δ	≈	≈	≈	≈	≈	Δ	≈	
f25	1.08E−02	4.36E−02	3.02E−11	3.40E−01	5.69E−01	4.12E−01	1.30E−01	1.19E−01	3.02E−11	9.07E−03	
	Δ	Δ	Δ	≈	≈	≈	≈	≈	Δ	Δ	
f26	4.12E−01	1.00E+00	6.72E−10	9.59E−01	6.79E−02	1.38E−02	7.39E−01	3.39E−02	3.08E−08	2.06E−01	
	≈	≈	Δ	≈	≈	Δ	≈	Δ	Δ	≈	
f27	6.84E−01	2.23E−01	1.55E−09	6.00E−01	5.49E−01	3.63E−01	9.63E−02	8.88E−01	6.05E−07	6.41E−01	
	≈	≈	Δ	≈	≈	≈	≈	≈	Δ	≈	
f28	3.11E−01	3.03E−02	3.02E−11	9.93E−02	1.91E−01	1.81E−01	3.95E−01	2.46E−01	3.02E−11	7.51E−01	
	≈	Δ	Δ	≈	≈	≈	≈	≈	Δ	≈	
f29	1.96E−01	2.97E−01	6.53E−07	7.24E−02	6.95E−01	5.11E−01	1.26E−01	8.77E−01	5.86E−06	9.12E−01	
	≈	≈	Δ	≈	≈	≈	≈	≈	Δ	≈	
f30	2.46E−01	3.39E−02	3.02E−11	2.90E−01	5.01E−01	9.59E−01	7.51E−01	3.18E−01	3.02E−11	4.64E−01	
	≈	Δ	Δ	≈	≈	≈	≈	≈	Δ	≈	
(Δ/∇/≈)	3/0/26	3/0/26	26/1/2	0/0/29	1/0/28	1/0/28	0/0/29	2/0/27	26/1/2	1/0/28	

Real-world engineering problems

In this section, five real-world engineering design problems are used to validate the search performance of each implemented algorithm. These problems are the speed reducer problem, tension compression spring design problem, welded beam design problem, pressure vessel, and three-bar truss design problem. All these problems are limited in nature, and therefore an external penalty approach mechanism was used to solve the design constraints. The maximum number of iterations for all problems was determined to be 1,000 and the population number was 30. The algorithm parameters are the values used in CEC’17. To visualize and compare the convergence behavior of the examined algorithms, the best fitness values obtained for each problem, which are usually called convergence curves, are drawn. All experiments corresponding to each algorithm were run independently 30 times. The best, mean, worst, and standard deviation values were examined comparatively, and the best solution obtained among the algorithms was highlighted in bold for ease of readability.

Speed reducer problem

This is essentially a gearbox issue, allowing the aircraft engine to spin at its most efficient speed (Dhiman, 2021). Finding the face width b (x1), the tooth modulus m ( x2), the number of teeth on the pinion z ( x3), the length of the first shaft between the bearings l1(x4), the length of the second shaft between the bearings l2(x5), the diameter of the first shaft d1(x6), and the diameter of the second shaft d2(x7) will allow you to determine the minimum values of the seven decision variables in this problem. Figure 5 shows a schematic illustration of the speed reducer concept. The goal of this design challenge is to determine the speed reducer’s lightest possible cost. The mathematical representation of this problem is as in the Appendix 1.

Figure 5 Speed reducer problem.

The comparative values of the performances of the 10 proposed CTSO methods and standard TSO methods on the speed reducer problem are presented in Table 6. The best, average, worst, and standard deviation values of the methods used are shown in Table 6. In addition, the decision variables depending on the best value of the methods used in the results of 30 runs on this problem are given in Table 7. When Table 6 is examined, it is seen that CTSO-3 and CTSO-9 are superior to other methods on the basis of the best value, and CTSO-9 is more successful when analyzed on the basis of mean value. In addition, the convergence graph of the methods used on the speed reducer problem is shown in Fig. 6.

Table 6 Statistical results of the used algorithms for speed reducer problem.

Algorithm	Best	Mean	Worst	SD	
TSO	3,043.966	338,765.8	2,180,277	634,222.4	
CTSO1	3,133.385	209,405.2	1,769,248	470,918.1	
CTSO2	3,008.135	643,475.8	2,297,437	856,143.2	
CTSO3	2,994.423	3,805.273	5,521.033	872.899	
CTSO4	3,050.252	422,165.5	2,001,481	630,221.1	
CTSO5	3,003.839	135,976.5	1,034,844	289,203.5	
CTSO6	3,015.958	231,801	2,363,101	636,178.5	
CTSO7	3,014.59	210,227.5	1,625,269	455,196.1	
CTSO8	3,050.025	594,289.3	2,401,982	850,915.1	
CTSO9	2,994.423	3,627.353	4,774.773	562.9862	
CTSO10	3,036.987	358,518.4	2,078,853	620,159.3	

Table 7 Comparison of the best optimum solution for the speed reducer problem.

Algorithm	Parameters values	f min	
x 1	x 2	x 3	x 4	x 5	x 6	x 7	
TSO	3.501019	0.7	17	7.3	7.917685	3.367619	5.349237	3,043.966	
CTSO1	3.499917	0.7	17	7.994088	7.911673	3.787018	5.28737	3,133.385	
CTSO2	3.504471	0.7	17	7.99709	7.857388	3.35903	5.287425	3,008.135	
CTSO3	3.49999	0.7	17	7.3	7.715319	3.350541	5.286655	2,994.423	
CTSO4	3.499871	0.7	17	8.067661	8.067661	3.361258	5.345938	3,050.252	
CTSO5	3.499944	0.7	17	8.006041	7.768105	3.353007	5.288778	3,003.839	
CTSO6	3.500284	0.7	17	8.048994	8.074678	3.35588	5.29533	3,015.958	
CTSO7	3.509419	0.7	17	7.630128	8.036515	3.373788	5.28744	3,014.590	
CTSO8	3.499876	0.7	17	8.014485	7.877796	3.352811	5.356255	3,050.025	
CTSO9	3.49999	0.7	17	7.3	7.715319	3.350541	5.286654	2,994.423	
CTSO10	3.500698	0.7	17	7.3	8.072679	3.361987	5.335593	3,036.987	

Figure 6 The convergence graph of the methods used on the speed reducer problem.

Tension compression spring design problem

The goal of the tension/compression spring design issue, as outlined by Arora (Arora, 2004), is to provide a spring design with the least amount of weight possible. Figure 7’s schematic illustration of this minimization issue shows some of its restrictions, including the cut-off voltage, ripple frequency, and minimal deviation. There are three choice variables in the tension/compression spring dilemma. These are the following: the wire diameter d ( x1), the average coil diameter D ( x2), and the number of active coils N ( x3). The mathematical representation of this problem is as in the Appendix 2.

Figure 7 Tension compression spring design problem.

The comparative values of the performances of the 10 proposed CTSO methods and standard TSO methods on the tension compression spring design problem are presented in Table 8. The best, average, worst, and standard deviation values of the methods used are shown in Table 8. In addition, the decision variables depending on the best value of the methods used in the results of 30 runs on this problem are given in Table 9. When Table 8 is examined, it is seen that CTSO-9 is superior to other methods on the basis of the best value, and CTSO-6 is more successful when analyzed on the basis of mean value. In addition, the convergence graph of the methods used on the tension compression spring design problem is shown in Fig. 8.

Table 8 Statistical results of the used algorithms for tension-compression spring design problem (Case 1).

Algorithm	Best	Mean	Worst	SD	
TSO	0.012673	0.013698	0.017631	0.001075	
CTSO1	0.012715	0.013655	0.017233	0.00081	
CTSO2	0.012702	0.013486	0.016016	0.000722	
CTSO3	0.01268	0.014229	0.020937	0.001895	
CTSO4	0.012784	0.013732	0.016824	0.000896	
CTSO5	0.012714	0.013537	0.015575	0.000668	
CTSO6	0.012739	0.013413	0.015612	0.000617	
CTSO7	0.01268	0.013579	0.015701	0.000774	
CTSO8	0.012683	0.013597	0.017302	0.000961	
CTSO9	0.012670	0.013483	0.017367	0.001136	
CTSO10	0.012714	0.013481	0.015773	0.000701	

Table 9 Comparison of the best optimum solution for the tension compression spring design problem.

Algorithm	Parameters values	f min	
x1	x2	x3	
TSO	0.05215	0.367888	10.66628	0.012673	
CTSO1	0.053121	0.391907	9.497097	0.012715	
CTSO2	0.050521	0.329228	13.11549	0.012702	
CTSO3	0.050809	0.335913	12.62168	0.01268	
CTSO4	0.054124	0.418139	8.436915	0.012784	
CTSO5	0.050561	0.330017	13.0706	0.012714	
CTSO6	0.053384	0.398798	9.208447	0.012739	
CTSO7	0.052278	0.370952	10.50679	0.01268	
CTSO8	0.05217	0.368334	10.65104	0.012683	
CTSO9	0.05119	0.344834	12.02127	0.012670	
CTSO10	0.053257	0.395514	9.333766	0.012714	

Figure 8 The convergence graph of the methods used on the tension compression spring design problem.

Welded beam design problem

The primary goal of the welded beam design challenge is to create a beam for the least amount of money while adhering to specific constraints (Varol Altay & Alatas, 2020). Figure 9 depicts a welded beam construction made up of beam A and the welding process needed to join it to item B. Five nonlinear inequality constraints and four choice variables make up the issue. The weld thickness, weld joint length, element width, and element thickness are represented by the design parameters h ( x1), l ( x2), t ( x3), and b ( x4), respectively. The mathematical representation of this problem is as in the Appendix 3.

Figure 9 Welded beam design problem.

The comparative values of the performances of the 10 proposed CTSO methods and standard TSO methods on the welded beam design problem are presented in Table 10. The best, average, worst, and standard deviation values of the methods used are shown in Table 10. In addition, the decision variables depending on the best value of the methods used in the results of 30 runs on this problem are given in Table 11. When Table 10 is examined, it is seen that CTSO-1 is superior to other methods on the basis of the best value, and CTSO-4 is more successful when analyzed on the basis of mean value. In addition, the convergence graph of the methods used on the welded beam design problem is shown in Fig. 10.

Table 10 Statistical results of the used algorithms for welded beam design problem.

Algorithm	Best	Mean	Worst	SD	
TSO	1.691099	2.765785	3.713832	0.681901	
CTSO1	1.680985	2.706922	3.696062	0.69335	
CTSO2	1.692163	2.630818	3.687721	0.754546	
CTSO3	1.909833	3.307382	5.11409	0.943303	
CTSO4	1.699917	2.416367	3.745437	0.663697	
CTSO5	1.760383	2.792383	3.786065	0.544282	
CTSO6	1.721776	2.423815	3.684699	0.621039	
CTSO7	1.74493	2.67853	3.994741	0.69217	
CTSO8	1.764811	2.587149	3.675218	0.590975	
CTSO9	1.807061	3.159026	6.243182	1.010093	
CTSO10	1.739581	2.665869	3.709204	0.646727	

Table 11 Comparison of the best optimum solution for welded beam design problem.

Algorithm	Parameters values	f min	
x1	x2	x3	x4	
TSO	0.198654	3.371947	9.112068	0.202756	1.691099	
CTSO1	0.200332	3.329221	9.164297	0.200695	1.680985	
CTSO2	0.195439	3.539739	9.191104	0.198922	1.692163	
CTSO3	0.235488	3.063879	8.008572	0.261938	1.909833	
CTSO4	0.202884	3.345369	9.074833	0.204389	1.699917	
CTSO5	0.146154	4.743619	9.193987	0.19883	1.760383	
CTSO6	0.197589	3.424587	8.970536	0.209319	1.721776	
CTSO7	0.161839	4.37461	9.210771	0.198758	1.74493	
CTSO8	0.220815	3.104299	8.68097	0.223645	1.764811	
CTSO9	0.202509	3.452743	8.545876	0.230036	1.807061	
CTSO10	0.200633	3.398656	8.875607	0.213808	1.739581	

Figure 10 The convergence graph of the methods used on the welded beam design problem.

Pressure vessel problem

The major goal of this issue is to optimize vessel formation, material use, and welding costs (He & Zhou, 2018). The objective function is constructed using four variables: shell thickness ( x1), head thickness ( x2), inner radius ( x3), and length ( x4), without accounting for vessel height. This issue contains four restrictions that must be met. Figure 11 depicts the pressure vessel design problem’s schematic structure. The mathematical representation of this problem is as in the Appendix 4.

Figure 11 Pressure vessel design problem.

The comparative values of the performances of the 10 proposed CTSO methods and standard TSO methods on the pressure vessel design problem are presented in Table 12. The best, average, worst, and standard deviation values of the methods used are shown in Table 12. In addition, the decision variables depending on the best value of the methods used in the results of 30 runs on this problem are given in Table 13. When Table 12 is examined, it is seen that CTSO-2 is superior to other methods on the basis of the best value, and CTSO-5 is more successful when analyzed on the basis of mean value. In addition, the convergence graph of the methods used on the pressure vessel design problem is shown in Fig. 12.

Table 12 Statistical results of the used algorithms for pressure vessel design problem.

Algorithm	Best	Mean	Worst	SD	
TSO	6,591.539	9,358.735	15,198.4	2,047.207	
CTSO1	6,934.041	9,290.626	12,997.45	1,833.564	
CTSO2	6,325.269	8,872.022	13,581.73	2,116.409	
CTSO3	6,720.31	18,807.36	55,057.06	10,642.19	
CTSO4	6,485.754	8,708.514	13,722.88	1,568.458	
CTSO5	6,479.417	8,662.113	12,317.9	1,380.715	
CTSO6	6,494.753	8,878.358	13,742.33	1,651.033	
CTSO7	6,782.928	8,763.569	12,770.58	1,438.112	
CTSO8	6,722.005	8,687.164	14,207.82	1,804.541	
CTSO9	7,836.811	17,484.35	38,598.81	8,019.801	
CTSO10	6,794.113	8,275.311	11,169.49	1,131.863	

Table 13 Comparison of the best optimum solution for pressure vessel design problem.

Algorithm	Parameters values	f min	
x1	x2	x3	x4	
TSO	1	0.5	49.9536	98.7225	6,591.539	
CTSO1	1.0625	0.5625	53.06296	75.76345	6,934.041	
CTSO2	0.875	0.5	45.277	140.8673	6,325.269	
CTSO3	1	0.5625	51.66779	85.64049	6,720.31	
CTSO4	1	0.5	51.01429	90.49689	6,485.754	
CTSO5	1	0.5	51.10119	89.90082	6,479.417	
CTSO6	0.9375	0.5	47.36856	120.7645	6,494.753	
CTSO7	1.0625	0.5625	54.90602	63.63404	6,782.928	
CTSO8	1	0.5625	51.64737	85.79055	6,722.005	
CTSO9	1.25	0.5	51.69956	85.4083	7,836.811	
CTSO10	1.0625	0.5625	54.77942	64.46914	6,794.113	

Figure 12 The convergence graph of the methods used on the pressure vessel problem.

Three bar truss design problem

In civil engineering, this issue is referred to as a structural optimization problem. By modifying the cross-sectional areas ( x1 and x2) while taking into consideration the stress (σ) on each of the truss members, Nowacki’s issue seeks to reduce the volume of the three-bar truss. These values’ possible value ranges are 0≤x1,x2≤1. The graphical illustration of this problem’s mathematical formulation is shown in Fig. 13. The mathematical representation of this problem is as in the Appendix 5.

Figure 13 Three bar truss design problem.

The comparative values of the performances of the 10 proposed CTSO methods and standard TSO methods on the three-bar truss design problem are presented in Table 14. The best, average, worst, and standard deviation values of the methods used are shown in Table 14. In addition, the decision variables depending on the best value of the methods used in the results of 30 runs on this problem are given in Table 15. When Table 14 is examined, it is seen that CTSO-9 is superior to other methods on the basis of the best value, and CTSO-10 is more successful when analyzed on the basis of mean value. In addition, the convergence graph of the methods used on the three-bar truss design problem is shown in Fig. 14.

Table 14 Statistical results of the used algorithms for three-bar truss design problem.

Algorithm	Best	Mean	Worst	SD	
TSO	263.8964	264.5518	270.4717	1.648298	
CTSO1	263.8987	264.8722	270.7144	2.06893	
CTSO2	263.9028	264.7003	270.7078	1.696716	
CTSO3	263.9005	266.5551	274.3699	2.902827	
CTSO4	263.8962	264.4986	270.7534	1.375953	
CTSO5	263.8963	264.5397	270.3816	1.46023	
CTSO6	263.9002	264.6418	270.7240	1.362083	
CTSO7	263.8966	264.6414	270.6517	1.388371	
CTSO8	263.9014	264.5663	270.3992	1.670418	
CTSO9	263.8957	266.2048	281.5610	3.621051	
CTSO10	263.8962	264.3218	270.7161	1.223462	

Table 15 Comparison of the best optimum solution for three-bar truss design problem.

Algorithm	Parameters values	f min	
x1	x2	
TSO	0.787491	0.411599	263.8964	
CTSO1	0.786632	0.414043	263.8987	
CTSO2	0.786469	0.41449	263.9028	
CTSO3	0.791316	0.40082	263.9005	
CTSO4	0.788856	0.407739	263.8962	
CTSO5	0.787869	0.410515	263.8963	
CTSO6	0.790911	0.401968	263.9002	
CTSO7	0.789892	0.404812	263.8966	
CTSO8	0.785835	0.416332	263.9014	
CTSO9	0.789298	0.40648	263.8957	
CTSO10	0.787728	0.41092	263.8962	

Figure 14 The convergence graph of the methods used on the three-bar truss design problem.

Analysis of CTSOs in real-world engineering problems with other metaheuristic optimization algorithms

The speed reducer problem has been solved by many different researchers in the literature with different metaheuristic optimization methods. The classical TSO and the proposed CTSO method were compared with the Moth Flame Optimization Algorithm (MFO) (Mirjalili, 2015), Weighted Superposition Attraction (WSA) (Baykaso, 2015), Grey Wolf Optimization (GWO) (Mirjalili, Mirjalili & Lewis, 2014), Artificial Acari Optimization (AAO) (Czerniak, Zarzycki & Ewald, 2017), Sine Cosine Algorithm (SCA) (Mirjalili, 2016), and Arithmetic Optimization Algorithm (AOA) (Abualigah et al., 2021a) methods in the literature. The best cost and related decision variables obtained by the proposed method and the methods proposed by other researchers for the speed reducer problem are presented in Table 16. When the results obtained were examined, it was observed that the proposed CTSO-3 and CTSO-9 method gave a much better result than the classical TSO and other competitive methods.

Table 16 Comparison of the speed reducer problem.

Algorithm	Parameters values	f min	
x1	x2	x3	x4	x5	x6	x7	
MFO (Mirjalili, 2015)	3.497455	0.700	17	7.82775	7.712457	3.351787	5.286352	2,998.941	
WSA (Baykaso, 2015)	3.500	0.7	17	7.3	7.8	3.350215	5.286683	2,996.348	
GWO (Mirjalili, Mirjalili & Lewis, 2014)	3.501	0.7	17	7.3	7.811013	3.350704	5.287411	2,997.820	
AAO (Czerniak, Zarzycki & Ewald, 2017)	3.4999	0.7	17	7.3	7.8	3.3502	5.2877	2,997.058	
SCA (Mirjalili, 2016)	3.521	0.7	17	8.3	7.923351	3.355911	5.300734	3,026.838	
AOA (Abualigah et al., 2021a)	3.50384	0.7	17	7.3	7.72933	3.35649	5.2867	2,997.916	
TSO	3.501019	0.7	17	7.3	7.917685	3.367619	5.349237	3,043.966	
CTSO-3	3.49999	0.7	17	7.3	7.715319	3.350541	5.286655	2,994.423	
CTSO-9	3.49999	0.7	17	7.3	7.715319	3.350541	5.286654	2,994.423	

There are many methods in the literature to solve the tension compression spring design problem. The classical TSO and the proposed CTSO method were compared with the HGSO, AAO, Whale Optimization Algorithm (WOA) (Mirjalili & Lewis, 2016), Harris Hawk Optimization (HHO) (Heidari et al., 2019), Chaotic Bird Swarm Algorithm (CMBSA) (Varol Altay & Alatas, 2020), Salp Swarm Algorithm (SSA) (Mirjalili et al., 2017), Cumulative Binomial Probability Particle Swarm Optimization (CBPPSO) (Agrawal & Tripathi, 2021), and Competitive Bird Swarm Algorithm (CBSA) (Wang, Deng & Duan, 2018) methods in the literature. The results obtained from this comparison are given in Table 17. Table 17 presents the best cost and relevant decision variables for the tension compression spring design problem. When the literature and the results of the experiments are examined, it is concluded that CTSO-9 gives better results than classical TSO and competitive methods in the literature.

Table 17 Comparison of tension compression spring design problem.

Algorithm	Parameters values	f min	
x1	x2	x3	
HGSO (Hashim et al., 2019)	0.0518	0.3569	11.2023	0.0127	
AAO (Czerniak, Zarzycki & Ewald, 2017)	0.0517	0.3581	11.2015	0.0127	
WOA (Mirjalili & Lewis, 2016)	0.0512	0.3452	12.004	0.0127	
HHO (Heidari et al., 2019)	0.0543	0.4239	8.2187	0.0128	
CMBSA (Varol Altay & Alatas, 2020)	0.0519	0.3618	11.000	0.0127	
SSA (Mirjalili et al., 2017)	0.0516	0.3547	11.4059	0.0127	
CBPPSO (Agrawal & Tripathi, 2021)	0.0512	0.3465	11.9097	0.0127	
CBSA (Wang, Deng & Duan, 2018)	0.0516	0.3566	11.2918	0.0127	
TSO	0.05215	0.367888	10.66628	0.012673	
CTSO-9	0.05119	0.344834	12.02127	0.012670	

Welded beam design problem has been solved by many different researchers in the literature with different metaheuristic optimization methods. The classical TSO and the proposed CTSO method were compared with the HGSO, MAOA, HHO, CMBSA, CBBSA, CBPPSO, Chaotic Grey Wolf Optimization (CGWO) (Kohli & Arora, 2018), Differential Big Bang-Big Crunch Algorithm (DBCA) (Prayogo et al., 2018), Sonar Inspired Optimization (SIO) (Tzanetos & Dounias, 2020), and Hybrid Genetic Algorithm-Ant Colony Optimization-Particle Swarm Optimization (H-GA-ACO-PSO) (Tam et al., 2019) methods in the literature. The best cost and relevant decision variables obtained by the proposed method and the methods proposed by other researchers for the welded beam design problem are presented in Table 18. When the results obtained were examined, it was observed that the proposed CTSO-1 method gave a much better result than the classical TSO and other competitive methods.

Table 18 Comparison of welded beam design problem.

Algorithm	Parameters values	f min	
x1	x2	x3	x4	
HGSO (Hashim et al., 2019)	0.2054	3.4476	9.0269	0.2060	1.7260	
MAOA (Altay, 2022a)	0.2057	3.4705	9.0366	0.2057	1.7246	
HHO (Heidari et al., 2019)	0.1956	3.7730	9.0307	0.2060	1.7501	
CMBSA (Varol Altay & Alatas, 2020)	0.2057	3.4702	9.0377	0.2057	1.7249	
CBPPSO (Agrawal & Tripathi, 2021)	0.2057	3.4704	9.0366	0.2057	1.7249	
CGWO (Kohli & Arora, 2018)	0.3439	1.8836	9.0313	0.2121	1.7255	
DBCA (Prayogo et al., 2018)	0.2057	3.4705	9.0366	0.2057	1.7249	
SIO (Tzanetos & Dounias, 2020)	0.3314	2.0174	9.0459	0.2088	1.7621	
H-GA-ACO-PSO (Tam et al., 2019)	0.2057	3.4705	9.0366	0.2057	1.7249	
TSO	0.198654	3.371947	9.112068	0.202756	1.6911	
CTSO-1	0.200332	3.329221	9.164297	0.200695	1.6810	

The pressure vessel problem has been solved by many different researchers in the literature with different metaheuristic optimization methods. The classical TSO and the proposed CTSO method were compared with the MAOA, CSMA, AAO, HHO, CMBSA, SSA, CBPPSO, H-GA-ACO-PSO, and Adaptive Reinforcement Learning based Bat Algorithm (ARLBAT) (Meng, Li & Gao, 2019) methods in the literature. The best cost of the proposed method and the methods suggested by other researchers for the pressure vessel problem and the relevant decision variables are shown in Table 19. When the results are examined, it is seen that the CSMA method gives a better result than the proposed CTSOs, classical TSO and other competitive methods.

Table 19 Comparison of pressure vessel problem.

Algorithm	Parameters values	f min	
x1	x2	x3	x4	
MAOA (Altay, 2022a)	0.7953	0.3931	41.2274	187.7371	5,914.48511	
CSMA (Altay, 2022c)	0.7778	0.3845	40.3207	199.9850	5,882.0851	
AAO (Czerniak, Zarzycki & Ewald, 2017)	0.8125	0.4375	42.0985	176.6366	6,059.7140	
HHO (Heidari et al., 2019)	0.8540	0.4329	44.0025	154.3888	6,543.4802	
CMBSA (Varol Altay & Alatas, 2020)	0.7780	0.3850	40.3200	200.0000	5,883.8610	
SSA (Mirjalili et al., 2017)	0.7807	0.3859	40.4707	197.9081	6,149.0232	
CBPPSO (Agrawal & Tripathi, 2021)	1.125	0.6250	62.9866	20.00000	6,952.7200	
H-GA-ACO-PSO (Tam et al., 2019)	0.8125	0.4375	42.0984	176.6366	6,059.7143	
ARLBAT (Meng, Li & Gao, 2019)	0.8125	0.4375	42.0984	176.6366	6,059.7143	
TSO	1	0.5	49.9536	98.7225	6,591.5390	
CTSO-2	0.875	0.5	45.277	140.8673	6,325.2690	

The three bar truss problem has been solved by many different researchers in the literature with different metaheuristic optimization methods. The classical TSO and the proposed CTSO method were compared with the AOA, SSA, CBPPSO, Cuckoo Search (CS) (Gandomi, Yang & Alavi, 2013), and mine blast algorithm (MBA) (Sadollah et al., 2013) methods in the literature. The best cost and relevant decision variables obtained by the proposed method and the methods proposed by other researchers for the three bar truss problem are shown in Table 20. When the literature and the results of the experiments are examined, it is concluded that CTSO-9 gives better results than classical TSO and competitive methods in the literature.

Table 20 Comparison of for three bar truss problem.

Algorithm	Parameters values	f min	
x1	x2	
AOA (Abualigah et al., 2021a)	0.79369	0.39426	263.9154	
SSA (Mirjalili et al., 2017)	0.78866541	0.408275784	263.8958	
CS (Gandomi, Yang & Alavi, 2013)	0.78867	0.40902	263.9716	
MBA (Sadollah et al., 2013)	0.7885650	0.4085597	263.8959	
TSO	0.787491	0.411599	263.8964	
CTSO-9	0.789298	0.40648	263.8957	

Feature selection

This section shows a comparison of performance between the suggested 10 distinct CTSO approaches and the state-of-the-art over ten different benchmark datasets using standard TSO and the state-of-the-art (Australian, ionosphere, spectheart, sonar, wine, heart, thyroid, tic-tac-toe, vehicle, krvskp). These datasets, whose descriptions are provided in Table 21, include a broad variety of attributes and come in various shapes and sizes. The purpose of this experiment is to assess the applicability of the suggested CTSO approaches for feature selection by analyzing their impact on benchmark datasets and comparing their performance to existing techniques. The experiment will explicitly look at how much better the performance of different datasets is when compared to the other suggested approaches. The efficiency of the recommended CTSO approaches is assessed using the accuracy measure, the standard deviation, and the number of features. The minimum, average, and maximum fitness values are also taken into account.

Table 21 Description of used benchmark’s datasets.

Datasets	Number of features	Number of instances	Classes	
Australian	14	690	2	
Ionosphere	34	351	2	
Spectheart	44	267	2	
Sonar	60	208	2	
Wine	13	178	3	
Heart	13	270	2	
Thyroid	21	7,200	3	
Tic-tac-toe	9	958	2	
Vehicle	18	846	4	
Krvskp	36	3,196	2	

The population size for this experiment was set at 10, and the maximum number of iterations was 100. The suggested 10 different CTSO results were contrasted with standard TSO. Tables 22 and 23 for each of the 10 benchmark datasets show the proposed CTSO strategy and all other techniques investigated, together with their minimum and average fitness values. When Table 22 is examined, the CTSO-9 method gave the best results in five of the 10 data sets and came first. It was followed by the CTSO-3 method, which gave the best results in four out of 10 datasets. When Table 23 is examined, CTSO-1, CTSO-3, CTSO-5, and CTSO-9 methods achieved the best results in 2 out of 10 data sets. CTSO-2 and CTSO-8 did not find good results on any dataset.

Table 22 Comparison of the CTSOs approaches based on the minimum of the fitness values.

	Australian	Ionosphere	Spectheart	Sonar	Wine	Heart	Thyroid	Tic-tac-toe	Vehicle	Krvskp	
CTSO-1	0.1739	0.0286	0.1509	0.0244	0.0286	0.1667	0.0174	0.1832	0.2722	0.0407	
CTSO-2	0.1522	0.0286	0.1509	0.0488	0.0286	0.1667	0.0194	0.1832	0.2663	0.0407	
CTSO-3	0.1522	0.0429	0.1132	0.0488	0.0286	0.1852	0.0174	0.1780	0.2485	0.0313	
CTSO-4	0.1522	0.0429	0.1509	0.0244	0.0286	0.1481	0.0194	0.1780	0.2899	0.0407	
CTSO-5	0.1667	0.0429	0.1321	0.0244	0.0286	0.1296	0.0181	0.1780	0.2663	0.0454	
CTSO-6	0.1667	0.0429	0.1698	0.0488	0.0000	0.1852	0.0194	0.1780	0.2722	0.0438	
CTSO-7	0.1522	0.0429	0.1509	0.0488	0.0286	0.1481	0.0167	0.1780	0.2840	0.0579	
CTSO-8	0.1594	0.0286	0.1698	0.0244	0.0286	0.1481	0.0201	0.1780	0.2840	0.0391	
CTSO-9	0.1594	0.0429	0.1321	0.0244	0.0000	0.1296	0.0167	0.1780	0.2781	0.0266	
CTSO-10	0.1594	0.0429	0.1698	0.0000	0.0286	0.1481	0.0181	0.1832	0.2899	0.0297	
TSO	0.1884	0.0714	0.2075	0.0976	0.0571	0.2407	0.0264	0.2042	0.3077	0.0704	
AO	0.2174	0.0857	0.2453	0.0976	0.0857	0.2407	0.0319	0.2304	0.3136	0.1017	
SMA	0.1739	0.0714	0.1887	0.0976	0.0857	0.2407	0.0278	0.2461	0.3254	0.0908	
WOA	0.1739	0.0571	0.1698	0.0488	0.0571	0.2037	0.0229	0.2042	0.2840	0.0595	

Table 23 Comparison of the CTSOs approaches based on the average of the fitness values.

	Australian	Ionosphere	Spectheart	Sonar	Wine	Heart	Thyroid	Tic-tac-toe	Vehicle	Krvskp	
CTSO-1	0.1768	0.0629	0.1811	0.0585	0.0486	0.2037	0.0238	0.2042	0.2888	0.0520	
CTSO-2	0.1768	0.0529	0.1736	0.0610	0.0429	0.2056	0.0245	0.2042	0.2970	0.0548	
CTSO-3	0.1674	0.0500	0.1717	0.0561	0.0486	0.2056	0.0238	0.1984	0.2905	0.0601	
CTSO-4	0.1804	0.0557	0.1755	0.0585	0.0514	0.1963	0.0244	0.1848	0.3024	0.0527	
CTSO-5	0.1754	0.0500	0.1792	0.0537	0.0429	0.1815	0.0234	0.2073	0.2911	0.0546	
CTSO-6	0.1848	0.0629	0.1811	0.0659	0.0286	0.1926	0.0235	0.2000	0.2941	0.0576	
CTSO-7	0.1833	0.0714	0.1736	0.0561	0.0343	0.2056	0.0228	0.1848	0.2959	0.0726	
CTSO-8	0.1717	0.0543	0.1849	0.0561	0.0486	0.2093	0.0221	0.1969	0.2994	0.0562	
CTSO-9	0.1696	0.0543	0.1604	0.0561	0.0429	0.1907	0.0214	0.1995	0.2929	0.0538	
CTSO-10	0.1899	0.0643	0.1792	0.0512	0.0486	0.1852	0.0224	0.1990	0.2970	0.0592	
TSO	0.2268	0.0829	0.2283	0.0976	0.0971	0.2667	0.0332	0.2209	0.3124	0.0889	
AO	0.2638	0.0929	0.2679	0.1073	0.1286	0.2722	0.0382	0.2613	0.3331	0.1668	
SMA	0.1826	0.0857	0.1943	0.1024	0.1114	0.2574	0.0334	0.2555	0.3385	0.1319	
WOA	0.2014	0.0643	0.1868	0.0585	0.0600	0.2185	0.0276	0.2105	0.2982	0.0714	

Tables 24 and 25 for each of the 10 benchmark datasets show the proposed CTSO strategy and all other techniques investigated, together with their average and maximum accuracy measures. When Table 24 is examined, WOA method achieved the best results in 4 out of 10 data sets and were ranked first according to the average accuracy measure value. The CTSO-1, CTSO-2, CTSO-3, CTSO-4, CTSO-5, CTSO-8, TSO, and AO methods did not produce a high value in any of the 10 datasets. When Table 25 is examined, the CTSO-9 method obtained the highest value in 5 out of 10 data sets according to the maximum accuracy metric. It is followed by CTSO-6 and CTSO-7, producing the highest value in 4 out of 10 datasets. The CTSO-2, CTSO-6, TSO, AO, SMA, and WOA methods could not gain superiority in any data set.

Table 24 Comparison of the CTSOs approaches based on the average of accuracy measure.

	Australian	Ionosphere	Spectheart	Sonar	Wine	Heart	Thyroid	Tic-tac-toe	Vehicle	Krvskp	
CTSO-1	78.1884	92.1429	79.0566	92.1951	93.7143	77.5926	96.9792	78.1152	68.8757	92.8326	
CTSO-2	78.7681	91.5714	78.1132	92.1951	95.1429	76.6667	97.3056	77.5916	69.6450	92.9421	
CTSO-3	76.5942	92.1429	78.4906	92.6829	93.7143	76.2963	97.5486	78.1152	70.0592	91.1111	
CTSO-4	77.5362	92.2857	79.8113	93.4146	93.1429	78.5185	97.2292	80.0524	68.9349	92.0344	
CTSO-5	79.6377	92.8571	78.8679	92.1951	95.1429	76.6667	97.5278	77.4346	69.4675	92.9890	
CTSO-6	78.9130	92.5714	80.1887	93.4146	95.7143	77.5926	97.6181	78.4817	69.8225	92.9890	
CTSO-7	79.1304	92.7143	77.9245	91.7073	95.4286	78.3333	97.4167	81.3613	69.3491	91.1894	
CTSO-8	78.9130	92.4286	77.7358	93.6585	94.5714	76.6667	97.5417	79.5812	68.9349	92.5509	
CTSO-9	79.8551	92.2857	79.4340	91.9512	94.5714	77.4074	97.6458	78.1152	69.2308	93.0673	
CTSO-10	79.3478	92.4286	80.1887	93.4146	94.5714	79.4444	97.4583	79.0052	69.1124	91.0798	
TSO	70.3623	89.5714	74.3396	89.5122	86.8571	71.8519	96.2778	73.9267	66.0947	86.0094	
AO	73.6232	90.7143	73.2075	89.2683	87.1429	72.7778	96.1806	73.8743	66.6864	83.3177	
SMA	81.7391	91.4286	80.5660	89.7561	88.8571	74.2593	96.6597	74.4503	66.1538	86.8075	
WOA	79.8551	93.5714	81.3208	94.1463	94.0000	78.1481	97.2361	78.9529	70.1775	92.8638	

Table 25 Comparison of the CTSOs approaches based on the maximum of accuracy measure.

	Australian	Ionosphere	Spectheart	Sonar	Wine	Heart	Thyroid	Tic-tac-toe	Vehicle	Krvskp	
CTSO-1	81.8841	95.7143	84.9057	95.1220	97.1429	81.4815	97.5000	80.1047	70.4142	95.9311	
CTSO-2	82.6087	94.2857	83.0189	95.1220	97.1429	81.4815	97.9861	81.6754	73.3728	94.9922	
CTSO-3	81.1594	94.2857	83.0189	95.1220	97.1429	79.6296	98.2639	82.1990	75.1479	93.5837	
CTSO-4	80.4348	94.2857	84.9057	95.1220	94.2857	85.1852	97.5000	82.1990	70.4142	94.3662	
CTSO-5	82.6087	95.7143	81.1321	95.1220	97.1429	85.1852	97.9167	79.5812	72.1893	94.9922	
CTSO-6	82.6087	94.2857	83.0189	95.1220	97.1429	81.4815	98.0556	81.1518	72.1893	95.1487	
CTSO-7	82.6087	95.7143	84.9057	95.1220	97.1429	85.1852	98.1944	82.1990	71.0059	92.6448	
CTSO-8	81.1594	95.7143	83.0189	97.5610	97.1429	85.1852	97.9861	81.1518	70.4142	96.0876	
CTSO-9	84.0580	95.7143	84.9057	95.1220	100.0000	83.3333	98.1944	82.1990	72.1893	94.8357	
CTSO-10	84.0580	92.8571	83.0189	97.5610	97.1429	85.1852	97.6389	81.1518	71.0059	93.8967	
TSO	72.4638	91.4286	77.3585	90.2439	91.4286	74.0741	97.2222	76.4398	68.0473	90.7668	
AO	78.2609	91.4286	75.4717	90.2439	91.4286	75.9259	96.8056	76.9634	68.6391	89.8279	
SMA	82.6087	92.8571	81.1321	90.2439	91.4286	75.9259	97.2222	75.3927	67.4556	90.9233	
WOA	82.6087	94.2857	83.0189	95.1220	94.2857	79.6296	97.7083	79.5812	71.5976	94.0532	

Table 26 presents a comparison of CTSO approaches and competitive metaheuristic methods according to selected features. When Table 26 is examined, it is seen that the SMA method chooses the least feature in 5 out of 10 data sets. In addition, it was seen that CTSO-9, which obtained the highest accuracy value in the majority of data sets, obtained the highest results by choosing an average number of features compared to other methods.

Table 26 Comparison of for three bar truss problem.

	Australian	Ionosphere	Spectheart	Sonar	Wine	Heart	Thyroid	Tic-tac-toe	Vehicle	Krvskp	
CTSO-1	5.7	16.6	21.1	29.6	6	5.7	10.5	5.7	9.4	19.6	
CTSO-2	5.9	15.8	22.5	29.9	6.6	5.8	10	5.3	9.5	19.4	
CTSO-3	6.5	14.4	20.2	30.3	6.8	5	9.9	5.5	8.1	18.6	
CTSO-4	6	16.5	21.3	28.9	6.1	5.2	10.2	6	9.2	19.3	
CTSO-5	5.6	17	21.8	29.1	6.5	6.7	9.6	5.6	8.7	19.5	
CTSO-6	6.1	15.8	21.2	29.9	5.6	5.8	9.9	5.5	8.7	20	
CTSO-7	5.8	15.4	20.2	32.2	6.1	5.6	10	6	9	19.5	
CTSO-8	6.1	15.5	21.4	28.6	6.1	6.1	10.5	6	8.6	18.5	
CTSO-9	6	16.4	21.6	29.4	6.2	5.7	10.3	5.5	9	19.2	
CTSO-10	6.1	15.6	20.1	28.4	5.6	5.4	10.4	5.6	8.2	19.2	
TSO	6.9	16.2	19.7	29.5	5.6	5.5	9.4	5.7	9.4	19.2	
AO	5.3	16.4	21	30.1	5.9	5.8	10.9	5.4	9.3	19.1	
SMA	1.1	7.3	1	29.7	5.7	3.5	7.4	6.7	9.3	20.3	
WOA	2.6	8.3	11.9	22.4	5.1	5.6	7.4	6	7.5	20.9	

Conclusion

TSO is a recently developed physics-based method that draws inspiration from the transient behavior of switched electrical circuits with storage components like inductance and capacitance. Our motivation was that this method is very new, tends to be stuck with local optimal solutions, has difficulties maintaining the balance between exploration and exploitation, and is not working to improve the performance of TSO. In this article, a number of chaotic maps have been incorporated in order to customize the TSO settings. By using chaotic maps with ergodic, irregular, and stochastic features in chaotic map TSO, it is aimed at avoiding local solutions more easily compared to the TSO method. In this way, the balance between exploration and exploitation has been achieved, and global convergence has increased. Ten distinct chaotic maps have been studied on twenty-nine different benchmark test functions, five different real-world engineering problems, and 10 different UCI standard datasets for feature selection. The findings demonstrated that the use of chaotic maps is capable of greatly enhancing TSO’s overall performance in a general sense. The approach that employs the Gauss and sinusoidal maps for supplying the values of parameter c1 has been determined to be the best chaotic TSO after being put through a series of comparisons with other chaotic TSOs. Gaussian and sinusoidal maps create a better balance between exploration and exploitation, which prevents the algorithm from being stuck in a local optimal state while it is in the process of optimizing its parameters. This is the reason for the enhanced performance. The results of the simulation demonstrated that Gaussian and Sinusoidal maps performed better than the standard TSO for the majority of the benchmark functions, that Sinusoidal maps performed better than the standard TSO for the majority of the real-world engineering problems, and that CTSOs generally suggested in the feature selection performed better than the standard TSO overall. According to the findings of the statistical analysis, customized algorithms clearly improve both the dependability of the global optimality and the quality of the outcomes.

Due to the fact that chaotic TSOs are relatively new, one potentially fruitful direction for future study would be to investigate how these algorithms perform in a parallel or distributed setting. The proposed CTSO method can also be applied promisingly in many different fields. Among them, many different fields such as machine learning, artificial intelligence, image processing, different real-world engineering problems, data mining, and big data are suitable for the application of algorithms. Like other proposed metaheuristic optimization methods, CTSO cannot guarantee that its solution will always equal the global optimum, due to its stochastic nature.

Appendix 1: speed reducer problem

Minimize:

f(x)=0.7854x22x1(14.9334x3−43.0934+3.3333x32)+0.7854(x5x72+x4x62)−1.508x1(x72+x62)+7.477(x73+x63)

Subject to:

g1(x)=−x1x22x3+27≤0

g2(x)=−x1x22x32+397.5≤0

g3(x)=−x2x64x3x4−3+1.93≤0

g4(x)=−x2x74x3x5−3+1.93≤0

g5(x)=10x6−316.91×106+(745x4x2−1x3−1)2−1100≤0

g6(x)=10x7−3157.5×106+(745x5x2−1x3−1)2−850≤0

g7(x)=x2x3−40≤0

g8(x)=−x1x2−1+5≤0

g9(x)=x1x2−1−12≤0

g10(x)=1.5x6−x4+1.9≤0

g11(x)=1.1x7−x5+1.9≤0

with bounds:

0.7≤x2≤0.8,17≤x3≤28,2.6≤x1≤3.6,5≤x7≤5.5,7.3≤x5,x4≤8.3,2.9≤x6≤3.9

Appendix 2: tension compression spring design problem

Minimize:

f(x)=(x3+2)x12x2

Subject to:

g1(x)=1−x23x371785x14≤0

g2(x)=4x22−x1x212566(x2x1−3x14)+15108x12≤0

g3(x)=1−140.45x1x23x3≤0

g4(x)=x1+x21.5−1≤0

with bounds:

0.05≤x1≤2.00,0.25≤x2≤1.30,2.00≤x3≤15.0

Appendix 3: welded beam design problem

Minimize:

f(x)=1.10471x12x2+0.04811x3x4(14.0+x2)

Subject to:

g1(x)=τ(x)−τmax≤0

g2(x)=σ(x)−σmax≤0

g3(x)=x1−x4≤0

g4(x)=1.10471x12+0.04811x3x4(14.0+x2)−5.0≤0

g5(x)=0.125−x1≤0

g6(x)=δ(x)−δmax≤0

g7(x)=P−Pc(x)≤0

where;

τ(x)=(τ′)2+(2τ′τ′′)x22R+(τ′′)2,τ′=60002x1x2,τ′′=MRJ,M=6000(14.0+x22),R=x224+(x1+x32)2

J={x1x22[x2212+(x1+x32)2]},σ(x)=504000x4x32,δ(x)=2.1952x4x33,Pc(x)=4.013Ex32x4636196(1−E4Gx328)

Pc(x)=4.013Ex32x4636196(1−E4Gx328)

τmax=13600psi,σmax=30000psi,δmax=0.25in,E=30×106psi,G=12×106psi

with bounds:

0.125≤x1≤2,0.1≤x4≤2 ve 0.1≤x2,x3≤10

Appendix 4: pressure vessel problem

Minimize:

f(x)=0.6224x1x3x4+1.7781x2x32+3.1661x12x4+19.84x12x3

Subject to:

g1(x)=−x1+0.0193x3≤0

g2(x)=−x2+0.00954x3≤0

g3(x)=−πx32x4−43πx33+1296000≤0

g4(x)=x4−240≤0

with bounds:

0.51≤x1,x2≤99.49,10≤x3,x4≤200

Appendix 5: three bar truss design problem

Minimize:

f(x)=l(x2+22x1)

Subject to:

g1(x)=x22x2x1+2x12P−σ≤0

g2(x)=x2+2x12x2x1+2x12P−σ≤0

g3(x)=1x1+2x2P−σ≤0

where

l=100,P=2,andσ=2

with bounds:

0≤x1,x2≤1

Supplemental Information

Supplemental Information 1 Code and Datasets.

Click here for additional data file.

Additional Information and Declarations

Competing Interests

Author Contributions

Data Availability

The authors declare that they have no competing interests.

Osman Altay conceived and designed the experiments, performed the experiments, analyzed the data, performed the computation work, prepared figures and/or tables, authored or reviewed drafts of the article, and approved the final draft.

Elif Varol Altay conceived and designed the experiments, performed the experiments, analyzed the data, performed the computation work, prepared figures and/or tables, authored or reviewed drafts of the article, and approved the final draft.

The following information was supplied regarding data availability:

Data and code are available at Zenodo and GitHub:

ALTAY, OSMAN, & VAROL ALTAY, ELİF. (2023). A novel chaotic transient search optimization algorithm for global optimization, real-world engineering problems and feature selection (1.0). Zenodo. https://doi.org/10.5281/zenodo.7992380.

https://github.com/osmanaltay1905/Chaotic-transient-search-optimization-algorithm.

The Australian, ionosphere, spectheart, sonar, wine, heart, thyroid, tic-tac-toe, vehicle and krvskp datasets are available at UCI:

- Australian dataset: https://archive.ics.uci.edu/ml/datasets/Statlog+%28Australian+Credit+Approval%29.

- Ionosphere dataset: https://archive.ics.uci.edu/ml/datasets/Ionosphere.

- Spectheart dataset: https://archive.ics.uci.edu/ml/datasets/SPECT+Heart.

- Sonar dataset: https://archive.ics.uci.edu/ml/datasets/Connectionist+Bench+%28Sonar%2C+Mines+vs.+Rocks%29.

- Wine dataset: https://archive.ics.uci.edu/ml/datasets/Wine.

- Heart dataset: https://archive.ics.uci.edu/ml/datasets/Statlog+%28Heart%29.

- Thyroid dataset: https://archive.ics.uci.edu/ml/datasets/Thyroid+Disease.

- Tic-tac-toe dataset: https://archive.ics.uci.edu/dataset/101/tic+tac+toe+endgame.

- Vehicle dataset: https://archive.ics.uci.edu/ml/datasets/Statlog+%28Vehicle+Silhouettes%29.

- Krvskp dataset: https://archive.ics.uci.edu/dataset/22/chess+king+rook+vs+king+pawn.

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
