# Peer review of "A novel chaotic transient search optimization algorithm for global optimization, real-world engineering problems and feature selection"

_PeerJ Computer Science, doi:10.7717/peerj-cs.1526_

## Round 0.1 · original submission · Major Revisions

Your paper has been reviewed by two reviewers who asked for revisions of the paper. Please revise the paper according to comments by reviewers, mark all changes in new version of the paper and provide cover letter with replies to them point to point.

Reviewer 1 ·

Basic reporting

1. English language should be checked. Certain grammar, syntax, and style errors need to be resolved.
2. Literature could be expended to include papers indicating the possible field of application of the newly developed metaheuristics.
3. The Introduction is not written well. It is more a literature review than an introduction. The Introduction should provide a problem background, identify the aim of the study, and highlight the main results, conclusions and contributions of the study.
4. Most of what is currently an Introduction should be moved to a separate section under the name “Literature review”. In addition, the authors should clearly point out the research gaps identified by reviewing the literature that they will try to cover in their study.
5. The authors should consider reducing the number of figures and tables.

Experimental design

1. The authors should consider building a graphical representation of the method to help the readers understand the concept more easily.

Validity of the findings

1. The paper does not have a proper discussion. Authors did not discuss how the results can be interpreted in perspective of previous studies. Discussion should clearly and concisely explain the significance of the obtained results in order to demonstrate the actual contribution of the article to this field of research, when compared with the existing and studied literature. In addition, the discussion should point out the theoretical and practical implications of the study.
2. Future research directions are weak. The authors should provide at least 3-5 solid future research directions that would be interesting to most of the Journal readesrhip.

Additional comments

1. There are certain technical issues:
a) There should be at least a couple of sentences between headings of different levels (e.g. between section 2 and sub-section 2.1).
b) It is highly unusual to have only one sub-section within a section (for example there is a sub-section 5.1 and no other sub-section). Is it truly necessary to have 5.1?
c) Figures 17 and 18 are not mentioned anywhere in the main text. All figures must be quoted somewhere in the main text.
d) Tables 5, 23, and 24 are not mentioned anywhere in the main text. All tables must be quoted somewhere in the main text.
e) Abbreviations should be defined at their first mention in the paper. For example, the abbreviation UCI is not defined in the abstract, CEC is used before it is defined in the main text, etc. Check the rest of the abbreviations.
f) The quality of the figures is extremely low. Some of them can hardly be understood since they are of so poor quality. For example, there is nothing one can conclude or understand based on Figure 10.
g) References are not formatted in the same manner. In addition, some of them are missing some important elements. Check and complete all references.

Reviewer 2 ·

Basic reporting

See the below comments.

Experimental design

See the below comments.

Validity of the findings

See the below comments.

Additional comments

Thank you for inviting me as a reviewer for this manuscript “A novel chaotic transient search optimization algorithm for global optimization, real-world engineering problems and feature selection”. The paper is written well and has enough contribution to be published in Computer Science. However, hoping to assist the authors in their research efforts, I provide several suggestions for improving the presented work:
. Abstract - The abstract completely needs to be rewritten. The current abstract only describes the general purposes of the article. It should also include the article's main (1) impact and (2) significance on decision making systems. Note that a good abstract should contain aim, methods, findings and recommendations. In addition, it should cover five main elements, introduction, problem statement, methodology, contributions and results.
. Introduction - You should begin with the problem, the gap, then propose the research question and just after that say what they want to do to address that.
Where is the gap? And you should clearly why it is a gap? Once again, if you say that it is a gap, then try to build a case for the gap.
. Why you have used TSO algorithm? Why not other heuristic algorithms like GA, SA, ABC, DRSO, Grey Wolf etc.? Discuss advantages of these algorithms.
. You should extend the literature review with application of heuristic algorithms and discuss them to show gap. Remove papers published before 2018. I suggest authors to read and discuss below interesting papers: Sadhu, T., Chowdhury, S., Shubham Mondal, Jagannath Roy, Chakrabarty, J., & Lahiri, S. K. (2022). A comparative study of metaheuristics algorithms based on their performance of complex benchmark problems. Decision Making: Applications in Management and Engineering. https://doi.org/10.31181/dmame0306102022r; Negi, G., Kumar, A., Pant, S., & Ram, M. (2021). Optimization of Complex System Reliability using Hybrid Grey Wolf Optimizer. Decision Making: Applications in Management and Engineering, 4(2), 241-256.; Mzili , T., Riffi , M. E., Mzili, I., & Dhiman, G. (2022). A novel discrete Rat swarm optimization (DRSO) algorithm for solving the traveling salesman problem. Decision Making: Applications in Management and Engineering, 5(2), 287-299.; Das, M., Roy, A., Maity, S., Kar, S., & Sengupta, S. (2022). Solving fuzzy dynamic ship routing and scheduling problem through new genetic algorithm. Decision Making: Applications in Management and Engineering, 5(2), 329-361. Madić, M., Gostimirović, M., Rodić, D., Radovanović, M., & Coteaţă, M. (2022). Mathematical modelling of the co2 laser cutting process using genetic programming. Facta Universitatis, Series: Mechanical Engineering, 20(3), 665-676.
. Provide more detail discussion on the results. A discussion section would allow you to come back to your research question and explain once again how their study inform literature in the proposed field in general. I suggest authors to provide some correlation discussion with results obtained based on other relevant algorithms in literature.
. I suggest authors to show benefits and limitations of your algorithm in comparisons to existing used for comparisons. This should be discussed.
. The conclusion section also seems to rush to the end. The authors will have to demonstrate the impact and insights of the research. The authors need to rewrite the entire conclusion section with focus on both impact and insights of the manuscript. Clearly state your unique research contributions in the conclusion section. No bullets should be used in your conclusion section. Provide some future directions.

---

## Round 0.2 · accepted · Accept

Dear authors,

Your revised version of the paper has been accepted. Both reviewers have accepted the paper.

Reviewer 1 ·

Basic reporting

All issues have been resolved.

Experimental design

All issues have been resolved.

Validity of the findings

All issues have been resolved.

Additional comments

All issues have been resolved.

Reviewer 2 ·

Basic reporting

The authors have addressed the point of my concern. I am happy with their corrections. Hence, I would like to recommend this manuscript to be published.

Experimental design

The authors have addressed the point of my concern. I am happy with their corrections. Hence, I would like to recommend this manuscript to be published.

Validity of the findings

The authors have addressed the point of my concern. I am happy with their corrections. Hence, I would like to recommend this manuscript to be published.

Additional comments

The authors have addressed the point of my concern. I am happy with their corrections. Hence, I would like to recommend this manuscript to be published.